# Entanglement of nanophotonic quantum memory nodes in a telecom network

C. M. Knaut[1,6], A. Suleymanzade[1,6], Y.-C. Wei[1,6], D. R. Assumpcao[2,6], P.-J. Stas[1,6], Y. Q. Huan[1], B. Machielse[1,3], E. N. Knall[2], M. Sutula[1], G. Baranes[1,4], N. Sinclair[2], C. De-Eknamkul[3], D. S. Levonian[1,3], M. K. Bhaskar[1,3], H. Park[1,5], M. Lončar[2] & M. D. Lukin[1✉]

A key challenge in realizing practical quantum networks for long-distance quantum communication involves robust entanglement between quantum memory nodes connected by fibre optical infrastructure[1–3]. Here we demonstrate a two-node quantum network composed of multi-qubit registers based on silicon-vacancy (SiV) centres in nanophotonic diamond cavities integrated with a telecommunication fibre network. Remote entanglement is generated by the cavity-enhanced interactions between the electron spin qubits of the SiVs and optical photons. Serial, heralded spin-photon entangling gate operations with time-bin qubits are used for robust entanglement of separated nodes. Long-lived nuclear spin qubits are used to provide second-long entanglement storage and integrated error detection. By integrating efficient bidirectional quantum frequency conversion of photonic communication qubits to telecommunication frequencies (1,350 nm), we demonstrate the entanglement of two nuclear spin memories through 40 km spools of low-loss fibre and a 35-km long fibre loop deployed in the Boston area urban environment, representing an enabling step towards practical quantum repeaters and large-scale quantum networks.

Distributing quantum entanglement between quantum memory nodes separated by extended distances[1,4] is an important element for the realization of quantum networks, enabling potential applications ranging from quantum repeaters[2,5] and long-distance secure communication[6,7] to distributed quantum computing[8,9] and distributed quantum sensing and metrology[10,11]. Proposed architectures require quantum nodes containing multiple long-lived qubits that can collect, store and process information communicated by photonic channels based on telecommunication (telecom) fibres or satellite-based links. In particular, the abilities to herald on successful photon arrival events and to detect quantum-gate errors are central to scalable implementations. As photons and individual matter qubits interact weakly in free space[12], a promising approach to enhance the interaction between light and communication qubits is to use nanophotonic cavity quantum electrodynamic (QED) systems, in which tight light confinement inside the nanostructure enables strong interactions between the photon and the communication qubit[13–16]. Furthermore, nanophotonic systems offer a path towards large-scale manufacturing and on-chip electric and optical control integration[17–19]. Several experiments demonstrated remote entanglement in systems ranging from neutral atoms[20–23] and trapped ions[24,25] to semiconductor quantum dots[26] and nitrogen-vacancy centres in diamond[27,28]. Recently, two atomic ensemble memories have been entangled through a metropolitan fibre network[29–31]. However, real-world applications require a combination of efficient photon coupling, long-lived heralded memory

and multi-qubit operations with practical telecom fibre networks, which is an outstanding challenge.

Here we report the realization of a two-node quantum network between two multi-qubit quantum network nodes constituted by silicon-vacancy (SiV) centres in diamond coupled to nanophotonic cavities and integrated with a telecom fibre network. SiVs coupled to cavities have emerged as a promising quantum network platform, having demonstrated memory-enhanced quantum communication[32] and robust multi-qubit single-node operation[33]. We extend these single-node experiments by demonstrating remote entanglement generation between two electron spins in two spatially separated SiV centres with a success rate of up to 1 Hz. Our approach uses serial, heralded spin-photon gate operations with time-bin qubits for robust entanglement of separated nodes and does not require phase stability across the link. We further make use of the multi-qubit capabilities to entangle two long-lived nuclear spins, using integrated error detection to enhance entanglement fidelities and dynamical decoupling sequences to extend the entanglement duration to 1 s. Both entanglement generation techniques rely on the strong light–matter interaction enabled by the coupling of SiV to the nanophotonic cavity. To demonstrate the feasibility of deployed quantum networks using our platform, we use bidirectional quantum frequency conversion (QFC) to convert the wavelength of the photonic qubits to telecom wavelengths. Building on recently demonstrated compatibility of our platform with bidirectional QFC[34,35], we demonstrate remote entanglement

[1]Department of Physics, Harvard University, Cambridge, MA, USA. [2]John A. Paulson School of Engineering and Applied Sciences, Harvard University, Cambridge, MA, USA. [3]AWS Center for Quantum Networking, Boston, MA, USA. [4]Department of Physics and Research Laboratory of Electronics, Massachusetts Institute of Technology, Cambridge, MA, USA. [5]Department of Chemistry and Chemical Biology, Harvard University, Cambridge, MA, USA. [6]These authors contributed equally: C. M. Knaut, A. Suleymanzade, Y.-C. Wei, D. R. Assumpcao, P.-J. Stas. ✉e-mail: lukin@physics.harvard.edu

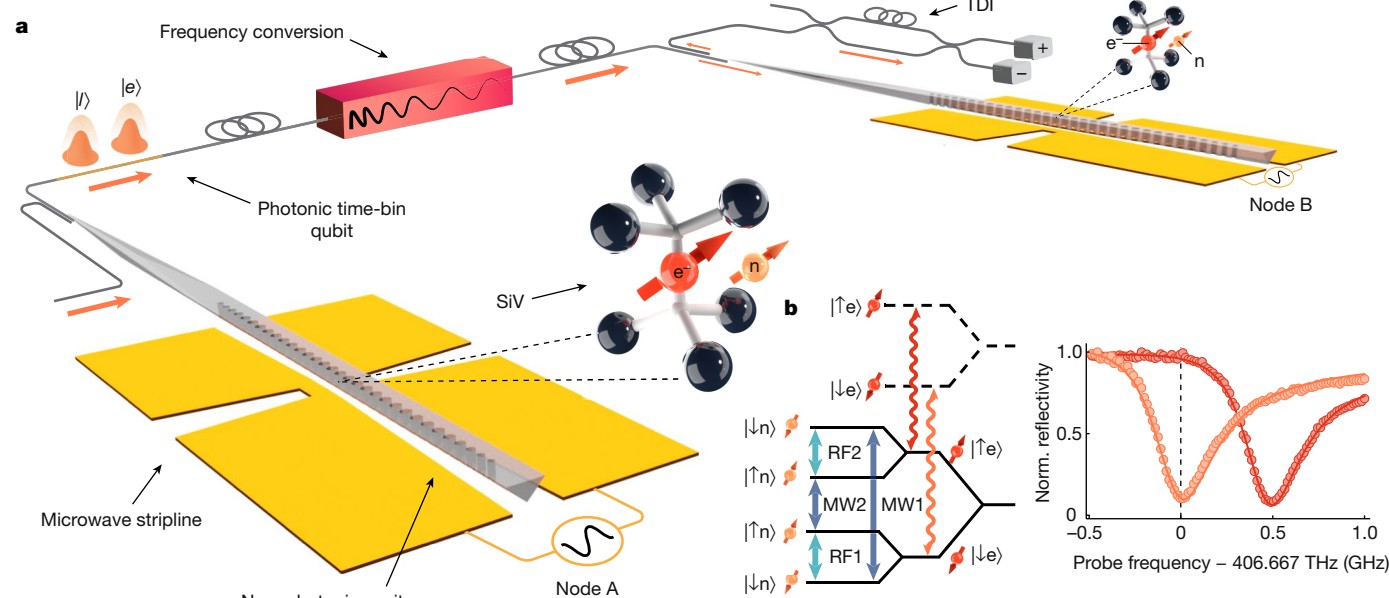

**Fig. 1 | A two-node quantum network of cavity-coupled solid-state emitters.**
**a**, Experimental setup. Each SiV is localized in a nanophotonic cavity within an individually operated cryostat held at temperatures below 200 mK in two separate laboratories. The line-of-sight distance between the two SiVs is 6 m. A gold coplanar waveguide is used to deliver microwave and radio-frequency pulses to the SiV. Both quantum network nodes are connected by an optical fibre of length $a \approx 20$ m and frequency-shifting setup to compensate for differences in the optical transition frequencies, or a long telecom fibre link using QFC (Fig. 4a). The measurement of the photonic time-bin qubit is performed at node B using a time-delay interferometer (TDI), which measures the time-bin qubit in the basis $|\pm\rangle \propto (|e\rangle \pm |l\rangle)$. **b**, Left, energy levels of $^{29}$SiV showing the microwave and radio-frequency transitions in the two-qubit manifold (blue and turquoise arrows) and the spin-conserving optical transitions (red and orange). Right, the reflection spectrum of cavity QED system of node A shows the electron-spin-dependent cavity reflectance. The dashed line indicates the frequency of maximum reflectance contrast, which is used as the frequency for the electron spin state readout and the photonic entanglement. Norm., normalized.

## Two-node quantum network using integrated nanophotonics

Our quantum network nodes consist of SiV centres in diamond that reside in individually operated dilution refrigerator setups in separate laboratories (Fig. 1a). By selectively implanting the $^{29}$Si isotope into the diamond substrate, each SiV deterministically contains two addressable spin qubits: one electron spin used as a communication qubit, which couples strongly to itinerant photons, and one long-lived $^{29}$Si nuclear spin, used as a memory qubit to store entanglement. Under an externally applied magnetic field, Zeeman sublevels define the electronic spin qubit states ($|\downarrow_e\rangle, |\uparrow_e\rangle$) and the nuclear spin qubit states ($|\downarrow_n\rangle$, $|\uparrow_n\rangle$) (refs. 36,37) (Fig. 1b, left). Microwave pulses are used to drive the electronic spin-flipping transitions, whereas radio-frequency pulses drive the nuclear spin-flipping transitions[33]. The SiV centres are embedded into nanophotonic diamond cavities, which enhance interactions between light and the electron spin[12,38]. The strong emitter–cavity coupling as characterized by the single-photon cooperativity in node A of 12.4 and node B of 1.5 (Supplementary Information) results in an electron-spin-dependent cavity reflectance[14] (Fig. 1b, right). This can be used to construct a reflection-based spin-photon gate (e–γ gate), which contains a sequence of rapid microwave gates generating entanglement between the electron spin of the SiV and the photonic qubits[14]. Moreover, taking advantage of the strong coupling between the electron spin of SiV and the $^{29}$Si nuclear spin, nucleus–photon entanglement can be created using the photon–nucleus entangling (PHONE) gate as demonstrated recently[33]. The two nodes are connected either directly by an optical fibre of length $a \approx 20$ m (Fig. 1a) or by a considerably longer telecom fibre link as discussed below (Fig. 4a).

generation through spools of up to 40 km of low-loss telecom fibre. Finally, we combine these techniques to demonstrate entanglement generation through a 35-km-long loop of fibre with 17 dB loss deployed in the Boston area urban environment.

We use a serial network configuration to generate remote entanglement between the electron spins in node A and node B, mediated by a time-bin photonic qubit (Fig. 2a). We first use a e–γ gate to generate an entangled Bell state between electron spin $|\downarrow_e^A\rangle, |\uparrow_e^A\rangle$ of node A and an incoming time-bin photonic qubit $|e\rangle, |l\rangle$ (ref. 14). Here, $|e\rangle$ and $|l\rangle$ describe the presence of a photon in the early and late time bins of the photonic qubit, which are separated by $\delta t = 142$ ns, respectively. The resulting photon–electron Bell state can be described as $|$Photon, SiV A$\rangle = (|e\downarrow_e^A\rangle + |l\uparrow_e^A\rangle)/\sqrt{2}$ (Methods). After that, the photonic qubit travels by optical fibre to node B, in which a second e–γ gate entangles the photonic qubit with the electron spin in node B. In the ideal, lossless case, the resulting state is a three-particle Greenberger–Horne–Zeilinger (GHZ) state:

$$|\text{Photon, SiV A, SiV B}\rangle = (|e\downarrow_e^A\downarrow_e^B\rangle + |l\uparrow_e^A\uparrow_e^B\rangle)/\sqrt{2}$$
$$= (|+\rangle|\Phi_{ee}^+\rangle + |-\rangle|\Phi_{ee}^-\rangle)/\sqrt{2}.$$

Here, $|\pm\rangle = (|e\rangle \pm |l\rangle)/\sqrt{2}$ describes two orthogonal superposition states of the photonic time-bin qubit, and $|\Phi_{ee}^\pm\rangle = (|\downarrow_e^A\downarrow_e^B\rangle \pm |\uparrow_e^A\uparrow_e^B\rangle)/\sqrt{2}$ describes the maximally entangled Bell states of the two spatially separated electron spins. The photonic qubit is measured in the $|\pm\rangle$ basis using a TDI to herald the generation of an electronic Bell state:

$$|\text{SiV A, SiV B}\rangle = \begin{cases} |\Phi_{ee}^+\rangle, & \text{if TDI measures } |+\rangle \\ |\Phi_{ee}^-\rangle, & \text{if TDI measures } |-\rangle. \end{cases}$$

Note that similar to the previously used single-node schemes[14], this method is robust to photon loss, as any losses of photons can be detected by a missing heralding event. Furthermore, the main advantage of our serial scheme is that both the early and late time bins of the photonic qubit travel through the same path, so no phase or polarization locking is necessary to guarantee high interference contrast at

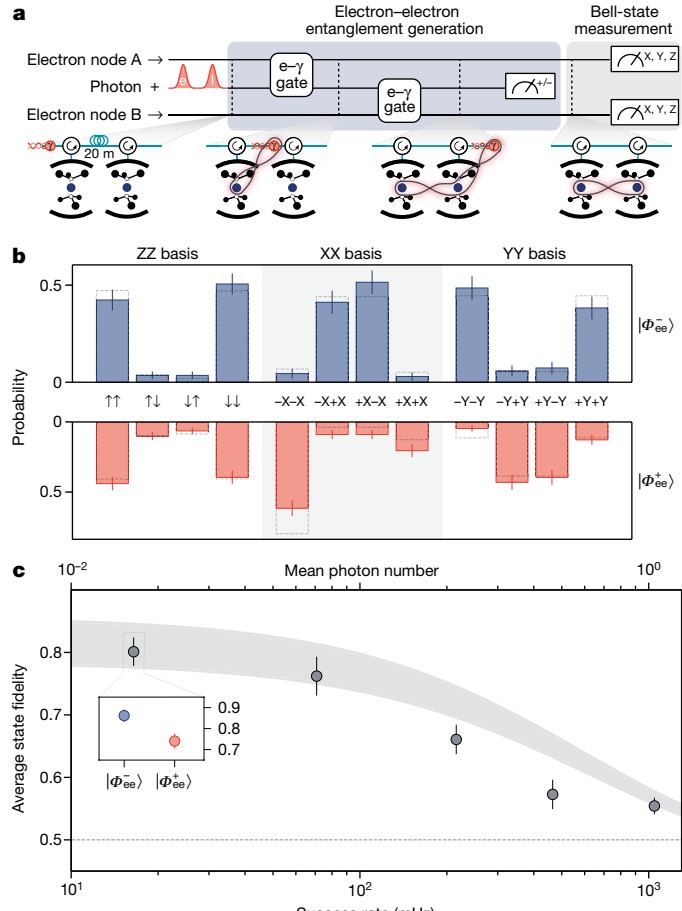

**b**

ZZ basis | XX basis | YY basis

$|\Phi_{ee}^-\rangle$

↑↑ ↑↓ ↓↑ ↓↓  −X−X −X+X +X−X +X+X  −Y−Y −Y+Y +Y−Y +Y+Y

Probability

$|\Phi_{ee}^+\rangle$

**c**

Mean photon number
$10^{-2}$ ... $10^0$

Average state fidelity

$|\Phi_{ee}^-\rangle$  $|\Phi_{ee}^+\rangle$

Success rate (mHz)
$10^1$ ... $10^2$ ... $10^3$

**Fig. 2 | Remote entanglement between two electronic spins. a**, Entanglement generation sequence. A photonic qubit is entangled with the electron spin in node A using the e–γ gate. A second e–γ gate entangles the photonic qubit with node B, generating a GHZ state among the two electronic qubits and the photonic qubit. A measurement of the photonic qubit in the |±⟩ basis heralds the generation of an electronic Bell state $|\Phi_{ee}^\pm\rangle$. **b**, Measurement results of Bell-state measurement. Measured correlations in the ZZ, XX and YY bases of the electronic spin corresponding to a Bell-state fidelity of $\mathcal{F}_{|\Phi_{ee}^-\rangle} = 0.86(3)$ (blue) and $\mathcal{F}_{|\Phi_{ee}^+\rangle} = 0.74(3)$ (red). Dashed bars show correlations predicted by a theoretical model using independently measured performance parameters of our system. **c**, Sweep of mean photon number of the photonic qubit showing that the success rates can be increased by sending photonic qubits with a higher mean photon number. The average fidelity of the generated $|\Phi_{ee}^+\rangle$ and $|\Phi_{ee}^-\rangle$ states is plotted. Inset, fidelities of states shown in **b**. Entanglement is shown to persist above the classical limit (dashed line) for success rates up to 1 Hz. Filled curves show predictions by a theory model using independently measured performance parameters of our system (Supplementary Information). Error bars in **b** and **c** are 1 s.d.

the TDI. This relaxes the requirements on system stability compared with one-photon schemes, which typically require an interferometric measurement of two emitted photons travelling through two stabilized paths[23,26,28,31] and avoids the reduction in entanglement rate typically present in two-photon schemes[27,39]. Furthermore, extending the number of network nodes to more than two can be achieved either by connecting more than two nodes in series or by using a switch network between multiple nodes to generate pairwise connectivity.

As cavity-coupled $^{29}$SiV centres possess an inhomogeneous distribution of optical transition frequencies of around ±50 GHz centred around 406.640 THz (737.2 nm), see ref. 40 and Methods, the frequency difference between the nodes needs to be coherently bridged. For node B used in this work, for instance, the optical frequency $\omega_B$ of the SiV is detuned from that of node A ($\omega_A$) by $\Delta_\omega = 13$ GHz. To address this, we

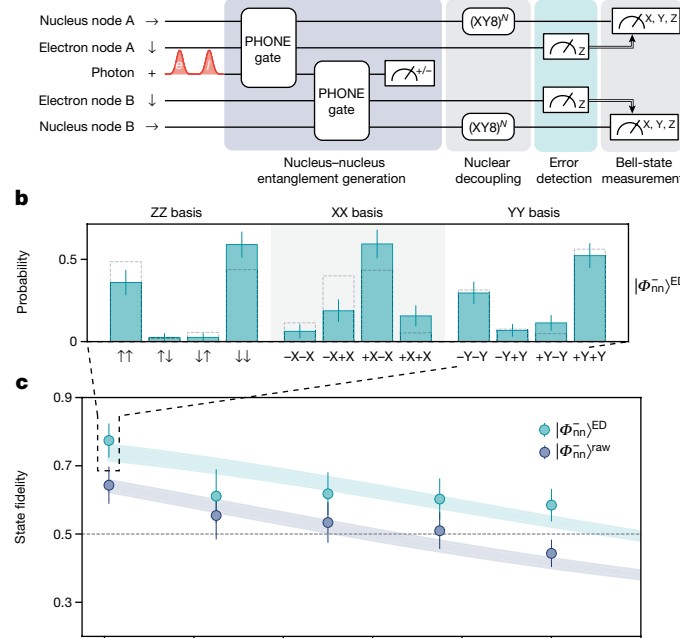

**b**

ZZ basis | XX basis | YY basis

$|\Phi_{nn}^-\rangle^{ED}$

↑↑ ↑↓ ↓↑ ↓↓  −X−X −X+X +X−X +X+X  −Y−Y −Y+Y +Y−Y +Y+Y

Probability

**c**

State fidelity

$|\Phi_{nn}^-\rangle^{ED}$
$|\Phi_{nn}^-\rangle^{raw}$

Total nucleus decoupling time (ms)
0 ... 200 ... 400 ... 600 ... 800 ... 1,000 ... 1,200

**Fig. 3 | Remote entanglement and long-lived storage using nuclear spins. a**, Entanglement generation and subsequent dynamical decoupling using nuclear spin qubits. Nuclear–nuclear entanglement is created by sequentially entangling a time-bin photonic qubit with the $^{29}$Si nuclei at nodes A and B using two PHONE gates. Measurement of the electron spin qubits allows for integrated error detection by flagging microwave gate errors that occurred during the PHONE gate. **b**, Results of Bell-state measurement of $|\Phi_{nn}^-\rangle$ after performing error detection, resulting in a Bell-state fidelity of $\mathcal{F}_{|\Phi_{nn}^-\rangle}^{ED} = 0.77(5)$. Dashed bars show correlations predicted by a theoretical model using independently measured performance parameters of our system. **c**, Decoherence protection of remotely entangled nuclear–nuclear Bell states, both with (turquoise) and without (blue) error detection. By performing XY8 dynamical decoupling sequences on the two nuclei, entanglement can be preserved for up to 1 s. Filled curves show predictions by a theory model using independently measured performance parameters of our system (Supplementary Information). The XY8-1 decoupling sequence was used for the datapoint with 10 ms decoupling time, whereas the XY8-128 sequence was used for all other measurements. The dashed line indicates the classical limit. Error bars in **b** and **c** are 1 s.d.

prepare the photonic qubit at frequency $\omega_A$ and then coherently shift its frequency by $\Delta_\omega$ after it has interacted with the SiV at node A, either using electro-optic frequency shifting or by bidirectional QFC[34,35].

### Electronic spin entanglement

To demonstrate the basic principles of network operation, we first focus on the nodes connected directly by an optical fibre of length $a \approx 20$ m and use electro-optical frequency shifting (see Methods for more details). The above protocol is applied using weak coherent states (WCS, with mean photon number $\mu = 0.017$) to encode time-bin qubits. After the TDI measurement heralds the generation of a Bell-state, single-qubit rotations and subsequent readout of the electron spin at each node implement the measurement of the correlations $\langle \sigma_i^A \sigma_i^B \rangle$, $i \in \{x, y, z\}$, which we abbreviate as XX, YY and ZZ, respectively. Figure 2b shows the results of the correlation measurements, from which we extract the fidelities of the resulting electron–electron state with respect to the maximally entangled Bell states $\mathcal{F}_{|\Phi_{ee}^-\rangle} = 0.86(3)$ (if the TDI measured |−⟩), and $\mathcal{F}_{|\Phi_{ee}^+\rangle} = 0.74(3)$ (if the TDI measured |+⟩), unambiguously demonstrating entanglement between the two nodes. The observed difference in fidelity is because of one source of infidelity associated with the imperfect reflection contrast of the two

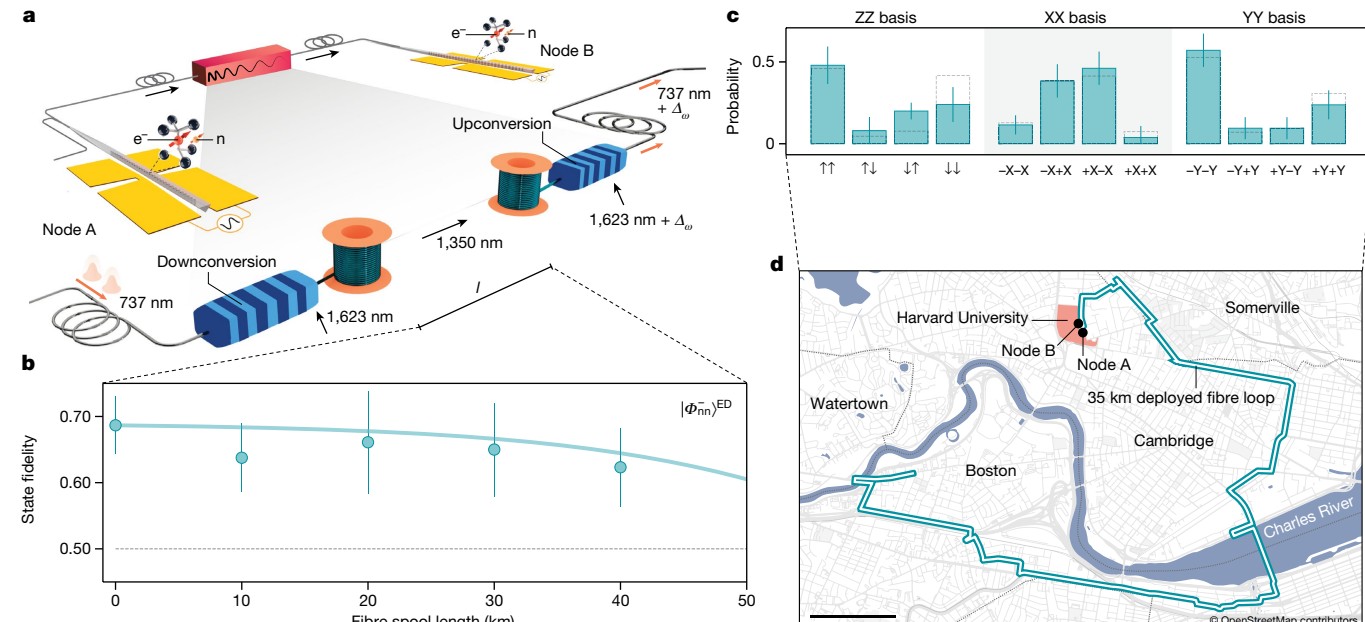

**Fig. 4 | Nuclear spin entanglement distribution through 35 km of deployed fibre. a**, Schematic of QFC setup. At node A, the photonic qubit is downconverted from 737 nm to 1,350 nm, which can propagate with low loss in telecom single-mode fibres. At the node B, it is upconverted back to 737 nm. The pump laser frequencies in the upconversion and downconversion setups are detuned by $\Delta_\omega = 13$ GHz to compensate for the difference in optical frequencies of the two SiVs. **b**, Nuclear spin Bell-state fidelities for varying lengths of telecom fibre spools between the two nodes. Entanglement persists for fibre lengths up to 40 km. Bell-state decoherence can be explained by a model incorporating a decrease in signal-to-noise ratio because of dark counts at 2.7 Hz and conversion noise photons at 2.5 Hz (solid line). The dashed line shows the classical limit. **c**, Measurement results of Bell-state measurement of $|\Phi_{nn}^-\rangle^{ED}$ state created through a 35-km long deployed fibre link shown in **d**, resulting in a fidelity of $\mathcal{F}_{|\Phi_{nn}^-\rangle}^{ED} = 0.69(7)$. Dashed bars show correlations predicted by a theoretical model using independently measured performance parameters of our system. **d**, Route of the deployed fibre link connecting nodes A and B. It consists of 35 km deployed telecom fibre routed towards and back from an off-site location, crossing four municipalities in the greater Boston metropolitan region. Error bars in **b** and **c** are 1 s.d. Scale bar, 1,000 m (**d**).

cavity-coupled SiVs. This results in reflection of the photonic qubit even when the electron spin is in the low-reflectivity $|\downarrow_e\rangle$ state. For our system configuration, this type of error accumulates preferentially for the $|\Phi_{ee}^+\rangle$ state, which is why $\mathcal{F}_{|\Phi_{ee}^+\rangle}$ is consistently lower than $\mathcal{F}_{|\Phi_{ee}^-\rangle}$ (Supplementary Information). Further error sources include contributions from 2− or higher photon number Fock states of the WCS used as time-bin photonic qubits. By varying the mean photon number $\mu$ in the WCS, we can increase the entanglement generation rate at the cost of reduced fidelity of the generated state. We explore this trade-off in Fig. 2c, in which we show that we are able to operate at success rates of 1 Hz while maintaining entanglement.

## Nuclear spin entanglement

Extending remote entanglement to larger distances requires the ability to preserve entanglement long enough such that the heralding signal obtained at node B can be classically relayed to node A. The coherence times of the electron spins in nodes A and B are 125 µs and 134 µs, respectively. Assuming classical communication using optical fibres in the telecom band, the decoherence of the electron spins would limit the distance between the nodes to approximately 25 km. To overcome this limitation, we demonstrate remote entanglement generation between two $^{29}$Si nuclei, which are long-lived quantum memories with storage times of more than 2 s (ref. 33). Analogous to the generation of electron–electron entanglement, remote nuclear entanglement is mediated by the photonic time-bin qubit (Fig. 3a). Thus, the first step of the remote entanglement generation sequence is creating entanglement between a photonic time-bin qubit and the $^{29}$Si nuclear spin at node A. This is achieved using the recently demonstrated PHONE gate, which uses only microwave pulses to directly entangle the $^{29}$Si nuclear spin with the photonic qubit (see ref. 33 and Methods), without the need to swap quantum information from electron to nuclear spin. After

applying the PHONE gate on the SiV in node A and the photonic qubit, in the ideal limit, their quantum state is

$$|\text{Photon, SiV A}\rangle = (|e \downarrow_n^A\rangle + |l \uparrow_n^A\rangle)\ |\downarrow_e^A\rangle/\sqrt{2}.$$

This implies that unless a microwave gate error has occurred, the electron spin is disentangled from the nuclear spin and is in the $|\downarrow_e^A\rangle$ state. Thus, the electron spin can be used as a flag qubit to perform error detection by discarding a measurement when the electron spin is measured in $|\uparrow_e^A\rangle$. By performing a second PHONE gate between the $^{29}$Si nuclear spin of node B and the time-bin qubit and by subsequently measuring out the photonic time-bin qubit in the $|\pm\rangle$ basis, the nuclear Bell states $|\Phi_{nn}^\pm\rangle$ are created. Following the entanglement generation, we perform XY8-type decoupling sequences on both nuclei to protect the nuclear–nuclear Bell state from decoherence caused by a quasi-static environment. Figure 3b shows the probability correlations of the resulting $|\Phi_{nn}^-\rangle$ state using a XY8-1 decoupling sequence with a total nuclear spin decoupling time of 10 ms. After using error detection by discarding measurements in which the electronic flag qubits are measured in the $|\uparrow_e\rangle$ state, the Bell-state fidelity is $\mathcal{F}_{|\Phi_{nn}^-\rangle}^{ED} = 0.77(5)$, which is an improvement from the directly measured value of $\mathcal{F}_{|\Phi_{nn}^-\rangle}^{raw} = 0.64(5)$ without error detection. Similar to $|\Phi_{ee}^+\rangle$, the generated $|\Phi_{nn}^+\rangle$ state accumulates errors because of imperfect reflectance contrast (Supplementary Information). Figure 3c shows Bell-state fidelities for longer total nuclear decoupling times. By performing XY8–128 decoupling sequences, entanglement can be preserved for up to 500 ms, with the application of error detection further extending this to 1 s.

## Entanglement distribution through 35 km of deployed fibre

Light at the resonant wavelength of the SiV (737 nm) experiences a high in-fibre loss of up to 4 dB km$^{-1}$, which limits the range of remote

entanglement distribution at this wavelength. To make our quantum network compatible with existing classical communication infrastructures that use low-loss optical fibres, we use bidirectional QFC to and from the telecom O-band (Fig. 4a); see the Methods. After the photonic qubit at 737 nm is reflected off the SiV of node A, a fibre-coupled PPLN waveguide pumped with 1,623 nm light converts the wavelength of the photonic qubit to 1,350 nm (ref. 34). This frequency lies in the telecom O-band and shows low attenuation (<0.3 dB km$^{-1}$) in conventional telecom single-mode fibre. After downconversion, the photonic qubit is sent through telecom fibre of varying length before a second PPLN upconverts the photonic qubit back to 737 nm. This bidirectional frequency conversion allows for straightforward bridging of the frequency difference $\Delta_\omega$ of the two SiVs: the frequency of the upconversion setup of the pump laser is offset by $\Delta_\omega$ from the frequency of the downconversion pump laser. The total efficiency of the bidirectional QFC, including a final filter cavity, is 5.4%, whereas the noise counts at the superconducting nanowire single-photon detector (SNSPD) of node B are 2.5 Hz.

Using this frequency conversion scheme together with the entanglement method described above (Fig. 3a), we remotely entangle two $^{29}$Si nuclei through spools of low-loss telecom fibre up to 40 km in length (Fig. 4b). For future repeater node applications of truly space-like separated quantum network nodes, it is important that entanglement persists until all nodes have received the classical heralding signal. To account for this effect, we execute an XY8–1 decoupling sequence for a total duration of 10 ms before performing the Bell-state measurement. The decoupling duration is much larger than the classical signal travelling time $\Delta t(l) \approx 200$ μs for the maximal fibre length of $l = 40$ km. Thus, for the measured fibre distances, Bell-state decoherence does not affect the measured Bell-state fidelities. Instead, we find that the fibre-distance-dependence of the nuclear–nuclear entanglement fidelities is well described by SNSPD dark counts and telecom conversion noise photons, which reduce the signal-to-noise ratio at high fibre attenuation (solid line in Fig. 4b).

In a practical setting, large-scale quantum networks can strongly benefit from existing fibre infrastructure to allow for long-distance entanglement distribution. Deployed fibres are subject to added noise and excess loss, as well as phase- and polarization drifts[34,35]. We demonstrate that our system is compatible with conventional fibre infrastructure and is resilient to these error sources by generating nuclear entanglement through a 35-km loop of telecom fibre deployed in the Boston area urban environment (Fig. 4d). The overall measured loss in the loop (17 dB at 1,350 nm) exceeds the nominal fibre attenuation of 11 dB at this wavelength, indicative of excess loss typical of deployed environments. As the input polarization of the upconverting PPLN needs to align with the dipole moment of the crystal, polarization drifts introduced by the deployed fibre are actively compensated to prevent a loss in conversion efficiency (Methods). Using the deployed link, we generate entanglement with a fidelity of $\mathcal{F}^{ED}_{|\Phi^-_{nn}\rangle} = 0.69(7)$ (Fig. 4c), demonstrating the quantum network performance in a realistic fibre environment.

## Outlook

Our experiments demonstrate key ingredients for building large-scale deployed networks using the SiV-based integrated nanophotonic platform. They open opportunities for exploration of a variety of quantum networking applications, ranging from distributed blind quantum computing[41] and non-local sensing, interferometry and clock networks[10,42], to the generation of complex photonic cluster states[43]. Extension to entanglement distribution between true space-like separated nodes using deployed fibre requires only relatively minor experimental modifications and is not limited by the performance of the quantum nodes (Supplementary Information). The success rate of the entanglement generation is currently limited by losses in the bidirectional QFC, which can be minimized by improving mode-matching into the PPLN and the efficiency of the filtering setup[44]. Furthermore, in-fibre attenuation could be further reduced to 0.2 dB km$^{-1}$ by using two-stage QFC to 1,550 nm (ref. 45). The use of WCS also reduces the success rate and fidelity, which could be avoided by using SiV-based single-photon sources[46] combined with active strain tuning of the nanophotonic cavities for wavelength matching[40,47]. Efficient coupling between the fibre network and the nanophotonic cavity could be improved by recently demonstrated cryogenic packaging techniques[48], whereas cooling requirements of the repeater nodes could be eased by deterministic straining of SiVs[49]. Entanglement fidelities could be improved by working with previously demonstrated nanophotonic cavities with higher cooperativity[32]. Implementing the above improvements, electron–electron entanglement fidelities of about 0.95 with success rates of about 100 Hz could be achieved (Supplementary Information). Finally, the number of accessible qubits could be increased by addressing weakly coupled $^{13}$C spins[50], allowing for more flexible multi-node network configurations. Combining these advances with the potential ability to create a large number of cavity QED systems fabricated on a chip, this approach can eventually result in large-scale, deployable quantum networking systems.

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

# Methods

## Spin-photon gates

The electron spin-dependent cavity reflectance is the building block of the photon–electron entangling gate (e–γ gate), and the photon–nucleus entangling (PHONE) gate. The functionality of the two gates can be described in a single-node configuration, in which a TDI measurement is performed after the node. For the e–γ gate (Extended Data Fig. 1a), the electron is initialized in the $|\text{SiV}\rangle = |{\rightarrow_e}\rangle = (|{\downarrow_e}\rangle + |{\uparrow_e}\rangle)/\sqrt{2}$ state, and a photonic time-bin qubit $|+\rangle = (|e\rangle + |l\rangle)/\sqrt{2}$ tuned to the frequency of the maximum reflectance contrast point between $|{\downarrow_e}\rangle$ and $|{\uparrow_e}\rangle$ is prepared and sent into the cavity-coupled SiV system. The early time bin is reflected only off the cavity QED system if the electron of the SiV is in the high-reflectivity state $|{\uparrow_e}\rangle$, and the state proportional to $|e{\downarrow_e}\rangle$ is traced out:

$$|\text{Photon, SiV}\rangle = |l\rangle(|{\downarrow_e}\rangle + |{\uparrow_e}\rangle)/\sqrt{3} + |e{\uparrow_e}\rangle/\sqrt{3}. \tag{1}$$

A NOT$_e$ pulse is applied to the electron between the two time bins, and the subsequent reflection of the late time bin results in a Bell state between the electron spin and the time-bin photonic qubit:

$$|\text{Photon, SiV}\rangle = (|e{\downarrow_e}\rangle + |l{\uparrow_e}\rangle)/\sqrt{2} \tag{2}$$

The spin-photon entanglement generation is heralded by the successful detection of a photon during the TDI measurement, such that single photons lost in transmission do not degrade the fidelity of the entangled state. The PHONE gate (Extended Data Fig. 1b) uses the electron spin to mediate entanglement between the nucleus and the photonic qubit. Before sending the time-bin qubit, the electron–nucleus register is initialized in the $|{\downarrow_e}{\rightarrow_n}\rangle$ state. A conditional electron spin-flipping microwave gate $C_n\text{NOT}_e$ then prepares the following electron–nucleus entangled state: $|\text{SiV}\rangle = (|{\downarrow_e}{\downarrow_n}\rangle + |{\uparrow_e}{\uparrow_n}\rangle)/\sqrt{2}$. The remaining sequence is similar to the e–γ gate, except with the addition of an unconditional NOT$_e$ pulse being applied to the electron between the two time bins. A final conditional electron spin-flipping microwave gate $\overline{C_n\text{NOT}_e}$ disentangles the electron and the nuclear spin and results in a Bell state between the nuclear spin and the time-bin photonic qubit:

$$|\text{Photon, SiV}\rangle = (|e{\downarrow_n}\rangle + |l{\uparrow_n}\rangle)|{\downarrow_e}\rangle/\sqrt{2}. \tag{3}$$

## Experimental setup

The optical path of the experiment consists of several stages shown in Extended Data Figs. 2 and 3. A laser table (Extended Data Fig. 2a) prepares all the required optical fields for control and feedback, as well as for the photonic qubits. The photonic qubit is sent to node A (Extended Data Fig. 2b), in which it interacts with the SiV–cavity system. The photonic qubit then continues to the frequency-shifting stage (Extended Data Fig. 3a) or telecom frequency conversion stage (Extended Data Fig. 3b) depending on the specific experiment, after which it travels to node B (Extended Data Fig. 2c) to interact with the second SiV–cavity system. Finally, the photonic qubit is measured in the $|+\rangle/|-\rangle$ basis with the TDI (Extended Data Fig. 2d). Each SiV at nodes A and B is located in a separate dilution refrigerator (BlueFors BF-LD250) with a base temperature below 200 mK. The SiV–cavity system is optically accessed through a tapered fibre. Lasers at 532 nm are used to stabilize the charge state of the SiVs at each node. The SNSPDs (Photon Spot) are used for SiV state readout, filter cavity locking and TDI locking. Counts from the SNSPDs are recorded on a time tagger (Swabian Instruments Time Tagger Ultra). A Zurich Instrument HDAWG8 2.4 GSa/s AWG is used for sequence logic, control of the acousto-optic modulators and phase electro-optic modulators (EOMs), as well as microwave control. The microwave and radio-frequency chain is identical for node A and B and is described in ref. 33.

## Frequency shifting

SiVs, similar to all solid-state qubits, show an inhomogeneous distribution in their transition frequencies because of their sensitivity to their local environment. In the case of the devices used in this work, we typically see a variation of roughly ±50 GHz for a SiV formed with a given Si isotope (Extended Data Fig. 4). In particular, the two SiVs used in this work have a difference in their optical transition frequency $\Delta_\omega = \omega_B - \omega_A = 13$ GHz, where $\omega_A$ is the frequency of the SiV in node A and $\omega_B$ is the frequency in node B.

To bridge this frequency difference and generate entanglement between the SiVs, electro-optic frequency shifting is used (Extended Data Fig. 3a). Briefly, the initial time-bin photon is generated at a frequency of $\omega_A$. After interacting with the SiV in node A, it is sent through a phase EOM, which is driven by a signal generator at a microwave frequency of $\Delta_\omega$. As the bandwidth of the optical photon is much narrower than $\Delta_\omega$ of the modulator, the frequency spectrum of the output photon can be approximated as a series of optical harmonic frequencies with intensity given by

$$I(\omega_A + k\Delta_\omega) \propto J_k^2(V_0/V_\pi) \tag{4}$$

where $k$ is an integer of the frequency harmonic used, $J_k$ is the $k$th Bessel function of the first kind, $V_\pi$ is the half-wave voltage and $V_0$ is the voltage applied to the modulator. As we are interested in shifting the output photon to a frequency $\omega_B$, which corresponds to the $k = 1$ harmonic of the output frequency spectrum, the power of the microwave drive of the EOM is set such that this harmonic is maximized. To filter out unwanted harmonics, the output of phase EOM is sent through a free space Fabry–Pérot cavity (linewidth = 160 MHz, finesse = 312) locked at frequency $\omega_B$. The overall efficiency is approximately 7.4%, given by the product of the maximum EOM sideband power occupancy (34%), the insertion loss of the EOM (about 50%) and the insertion loss of the filter cavity, including transmission through the cavity and free space to fibre losses (about 40%). We note that this method can be used to shift photons across much larger frequency differences without any additional degradation of efficiency, limited only by the bandwidth of practically available microwave sources and EOMs, and even beyond that using higher harmonics of the output frequency spectrum at the expense of efficiency.

## Telecom frequency conversion

To operate over long distances, it is beneficial to transmit photons over lower-loss telecom frequencies. As SiVs have a natural optical transition at 737 nm, which has an in-fibre loss of up to 4 dB km$^{-1}$, we use QFC to convert it to 1,350 nm, which is in the telecom O-band and has a loss of less than 0.3 dB km$^{-1}$ in SMF28 fibre.

The conversion was performed using fibre-coupled, reverse photon exchange lithium niobate waveguide devices (AdvR) with a pump at 1,623 nm. For downconversion 737 nm + 1,623 nm → 1,350 nm, a difference frequency generation (DFG) device was used with a pump power of 23 dBm at saturation, resulting in an overall 33% conversion efficiency. The upconversion 1,350 nm + 1,623 nm → 737 nm used a Sum Frequency Generation (SFG) device with a pump power of 25 dBm at saturation, resulting in an overall 30% conversion efficiency. The conversion efficiency in both directions is limited by impedance mismatch between the fibre and waveguide modes. The same pump laser is used for DFG and SFG processes with a frequency offset created by an EOM to compensate for frequency differences $\Delta_\omega = 13$ GHz of the SiVs in two nodes. A custom-built Fabry–Pérot cavity (linewidth = 160 MHz, finesse = 312) is used to filter excess noise from the pump at the output of the SFG device. For more details about the telecom setup, see Extended Data Fig. 3b.

## Polarization stabilization

In-field deployed fibres experience polarization drifts, which need to be compensated because of the polarization sensitivity of the

upconverting PPLN used after the deployed fibre link. Extended Data Fig. 5a shows the setup used to characterize and stabilize the change in the degree of polarization (DOP) introduced by the deployed fibre. At the input of the fibre link, a classical signal from a linearly polarized laser at 1,350 nm is sent through the deployed link. At the output of the link, a polarimeter measures the DOP as expressed by the two degrees of freedom ellipticity $\chi \in (-\pi/4, \pi/4)$ and azimuth $\psi \in (-\pi/2, \pi/2)$. Optical switches are used to connect the deployed fibre link to either the full experiment or the polarization stabilization setup. Extended Data Fig. 5b shows the histogram of measurements of $\chi$ and $\psi$ over a period of 5 days, without polarization stabilization, showing that the DOP traces out the full phase space over this period. To stabilize the polarization at the output of the fibre link to a well-defined DOP (arbitrarily chosen to be $\chi = \psi = 0$), a fibre-squeezer-based polarization controller is used. During the polarization stabilization sequence, the value of the following cost function is evaluated:

$$C(\chi, \psi) = \sqrt{(\cos(\chi) - 1)^2 + (\cos(\psi) - 1)^2} \qquad (5)$$

By using a gradient-descent algorithm, the optimal control voltages of the polarization controller are determined. The results of this polarization stabilization are shown in Extended Data Fig. 5c, which shows a histogram of the DOP measured during 5 days of nuclear–nuclear entanglement generation using the deployed fibre. The polarization stabilization routine is interleaved with the entanglement generation experiment and runs during 3% of the experimental time, minimally affecting the overall duty cycle. The insertion loss of the polarization stabilization setup is approximately 1.5 dB.

## Success rate

The success rate $r_{suc}$ of the heralded entanglement at the TDI can be described as $r_{suc} = \eta R D$, where $\eta$ is the success probability of a heralding event, $R$ is the repetition rate of the experiment and $D$ is the duty cycle of the experiment. The success probability is limited by the photonic link efficiency and protocol-specific loss channels, such as the mean photon number $\mu < 1$ of the WCS of the photonic qubit (Supplementary Information). The repetition rate $R$ is limited by the readout time for the electron spin of the two nodes, which is 67 μs and 17 μs for nodes A and B, respectively. The duty cycle $D$ is reduced from its ideal value of 1 by periodic locking of the TDI and the filter cavity after the frequency shifting or telecom conversion setup, active tracking of the optical frequency of SiV, and, in case of the entanglement generation experiment through the deployed fibre link, polarization stabilization. Furthermore, the application of the green ionization laser to reset the charge state of SiV further reduces $D$. By discarding measurements for which the spin-dependent reflectance contrast falls below a certain threshold, indicative of large spectral diffusion, $D$ is further reduced. On average, this contrast threshold for the SiV in node A was set to 1:16, whereas for the SiV in node B it was set to 1:8. These thresholds resulted in an average rejection of 23% of data points per dataset. Extended Data Table 1 shows the average values for $R$, $\eta$, $D$ and $r_{suc}$ for electron–electron entanglement generation and nuclear–nuclear entanglement generation.

## Data availability

All data related to this study are available from the corresponding author upon request.

## Code availability

All analysis codes related to this study are available from the corresponding author upon request.

**Acknowledgements** We thank D. Sukachev and E. Bersin for their discussions and experimental help; J. Borregaard, D. Englund, S. Guha and P. B. Dixon for their discussions; and J. MacArthur for assistance with electronics. This work was supported by the research alliance of the AWS Center for Quantum Networking with the Harvard Quantum Initiative, the National Science Foundation (NSF, grant no. PHY-2012023), NSF EFRI ACQUIRE (5710004174), CUA (PHY-2317134), AFOSR MURI (FA9550171002 and FA95501610323) and CQN (EEC-1941583). The devices were fabricated at the Harvard Center for Nanoscale Systems (NSF award no. 2025158). Y.Q.H. acknowledges support from the A*STAR National Science Scholarship. D.R.A. and E.N.K. acknowledge support from an NSF GRFP no. DGE1745303. M.S. acknowledges funding from the NASA Space Technology Graduate Research Fellowship Program. G.B. acknowledges funding from the MIT Peskoff Graduate Research Fellowship.

**Author contributions** C.M.K., A.S., Y.-C.W., D.R.A., P.-J.S., Y.Q.H., M.S., D.S.L., M.K.B. and M.D.L. planned the experiment. B.M. and E.N.K. fabricated the devices. C.M.K., A.S., Y.-C.W., D.R.A., P.-J.S. and Y.Q.H. built the setup and performed the experiment. C.M.K., A.S., Y.-C.W., D.R.A., P.-J.S., Y.Q.H., M.S. and G.B. analysed the data and interpreted the results. N.S. and M.K.B. coordinated the commissioning of the deployed fibre link. C.D.-E. provided support with the tapered-fibre-optical interface. All work was supervised by H.P., M.L. and M.D.L. All authors discussed the results and contributed to the paper.

**Competing interests** The authors declare no competing interests.

**Additional information**
**Correspondence and requests for materials** should be addressed to M. D. Lukin.

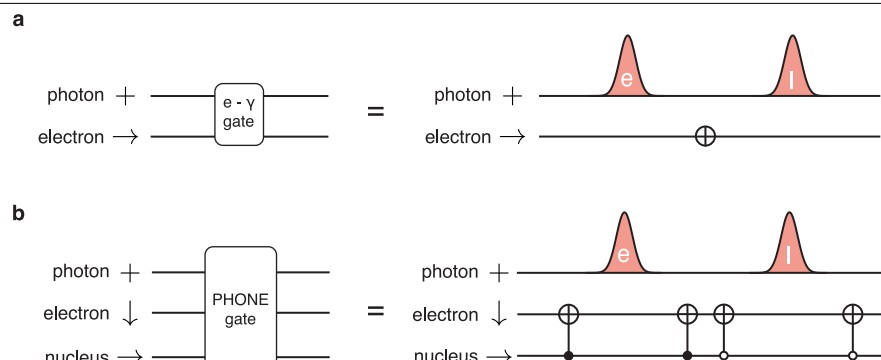

**Extended Data Fig. 1 | Spin-photon gate quantum circuits. a**, Gate representation of the *e-γ* gate, entangling a time-bin photonic qubit with the SiV's electron spin. **b**, Gate representation of the PHONE gate, entangling a time-bin photonic qubit with the SiV's $^{29}$Si nuclear spin.

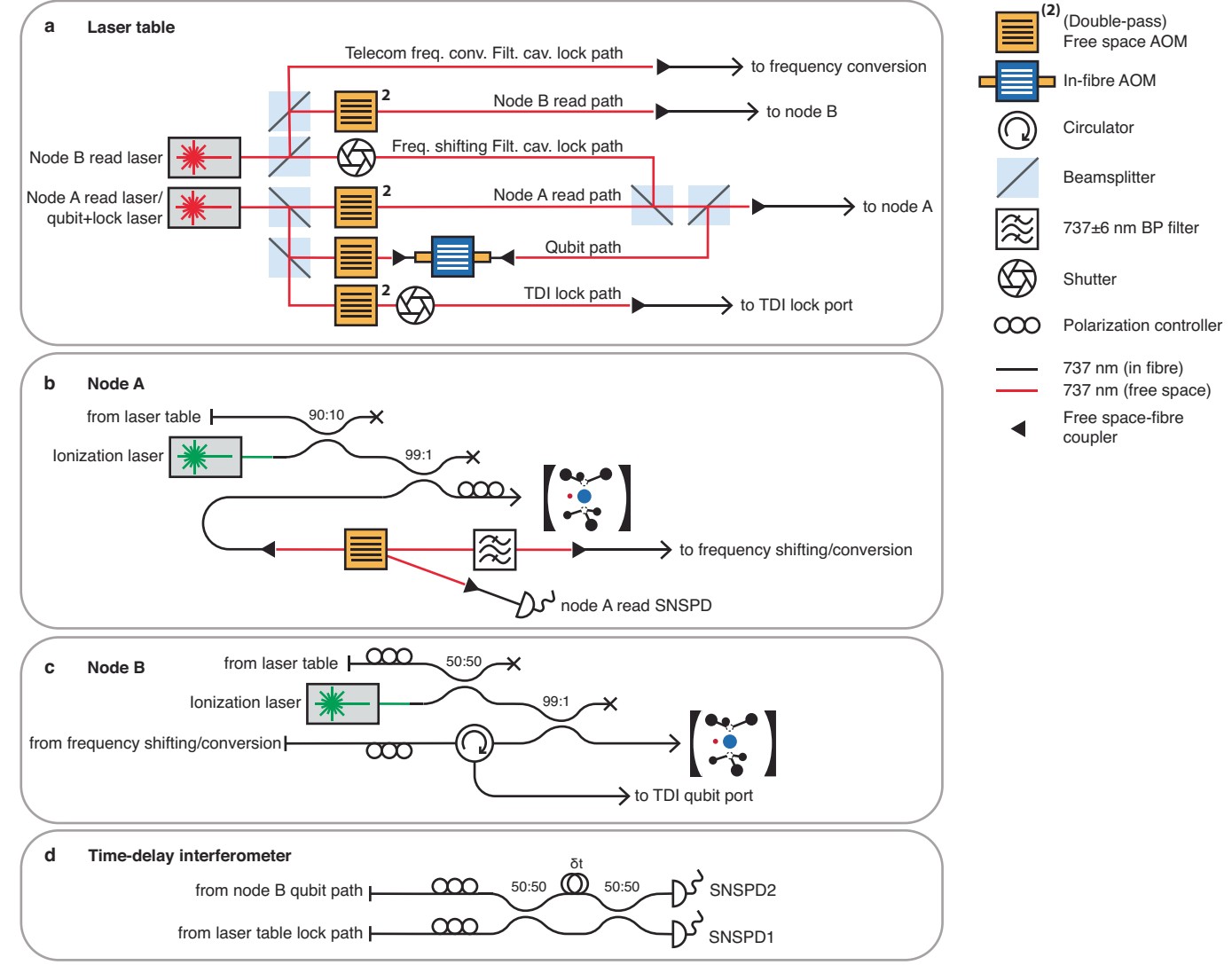

**Extended Data Fig. 2 | Schematic drawing of the two-node quantum network. a**, Laser table. Here, all the optical fields are prepared for node A and B readout, filter cavity, and time-delay interferometer (TDI) locking, and the photonic qubits are shaped by an in-fibre AOM. **b**,**c**, Node A and B. The nodes contain the photonic qubit travel path and a readout path for individual spin readout, as well as an insertion port for a green laser stabilizing the SiVs' charge state. **d**, Time-delay interferometer. The TDI allows measurement of the photonic qubits in superposition bases.

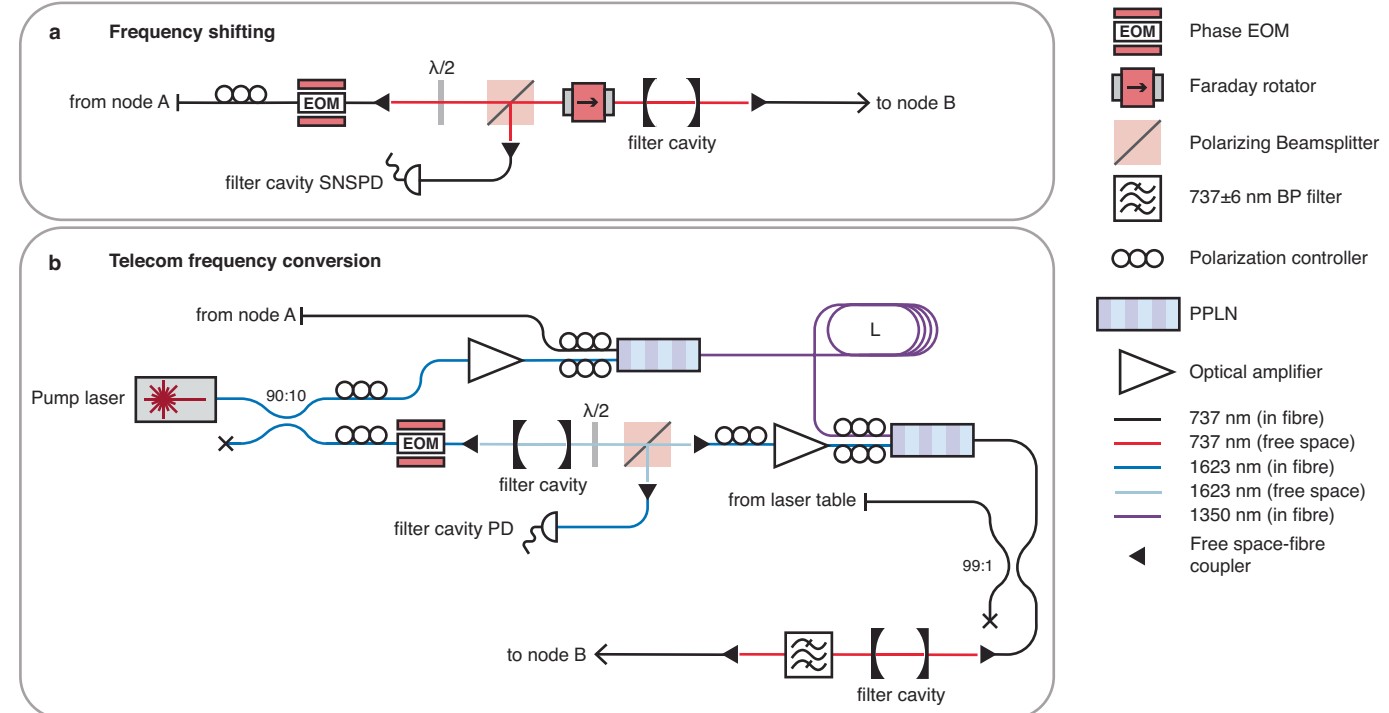

**Extended Data Fig. 3 | Frequency shifting/conversion setup. a**, Frequency shifting. This setup is used to directly bridge the frequency difference between the two nodes while staying in the visible light range. **b**, Telecom frequency conversion. This setup is used for long-distance entanglement generation. It contains two frequency conversion stages: one from visible to telecom, and one from telecom to visible. The two stages are separated by a variable amount of fibre spools of total length $L$, or a field-deployed 35 km fibre.

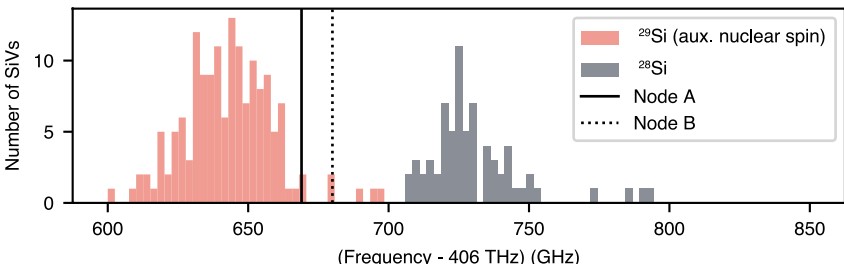

**Extended Data Fig. 4 | Optical resonance frequencies of 223 characterized cavity-coupled SiVs.** The results of this work were produced with SiVs with the $^{29}$Si isotope of silicon, which have a deterministic auxiliary nuclear spin memory. The optical resonance frequencies of the specific SiVs used in this work are given by the solid (node A) and dashed (node B) vertical lines.

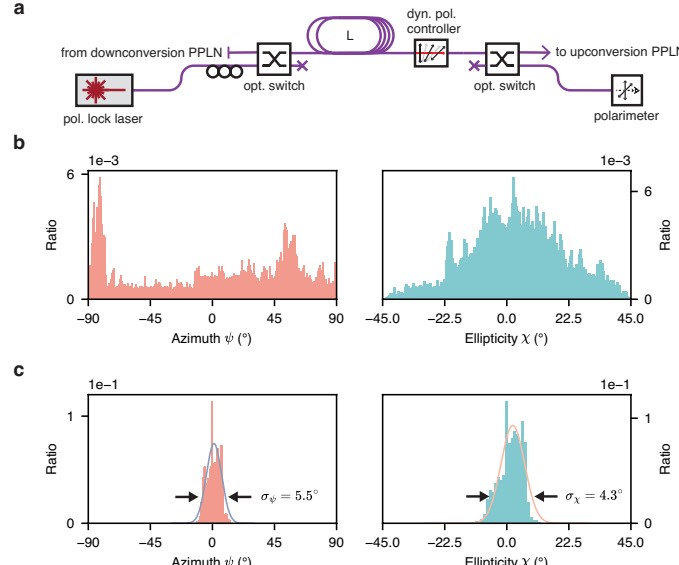

**Extended Data Fig. 5 | Polarization stabilization. a**, Experimental setup used to measure and stabilize the change of the DOP introduced by the 35 km fibre link. A linearly polarized lock laser can be launched into the fibre link. After the fibre link, the DOP is measured using a polarimeter. A fibre-based polarization controller is used to stabilize the DOP. **b**, DOP measurement during 5 hours through the 35 km fibre link, not using polarization stabilization. During the measurement period, the initially linearly polarized light traces out the full phase space of the DOP. **c**, DOP during 5 days of nuclear-nuclear entanglement generation, with polarization stabilization enabled. The DOP can be stabilized at around $\chi = \psi = 0$.

**Extended Data Table 1 | Success rates**

|  | Electron-Electron Entanglement | Nucleus-Nucleus Entanglement (ED) |
|---|---|---|
| Repetition Rate R | 10 kHz | 1.4 kHz |
| Success Probability $\eta$ | $7.7 \times 10^{-6}$ - $2.5 \times 10^{-4}$ | $2.0 \times 10^{-5}$ |
| Duty Cycle D | 34 % | 20 % |
| Success rate $r_{suc}$ | 16 mHz - 1050 mHz | 6 mHz |

Average values for repetition rate, success probability, and duty-cycle for electron-electron entanglement generation, shown in Fig. 2, and nucleus-nucleus entanglement generation, shown in Fig. 3. For the nucleus-nucleus entanglement generation, values have been averaged across all decoupling times and $|\Phi_{nn}^{-}\rangle^{ED}$ and $|\Phi_{nn}^{+}\rangle^{ED}$. For the electron-electron entanglement generation, the success probabilities vary owing to varying mean photon numbers in the photonic qubit.