## [Peer Review File · Nature]

Manuscript Title: Entanglement of Nanophotonic Quantum Memory Nodes in a Telecom Network

Reviewer Comments & Author Rebuttals

Reviewer Reports on the Initial Version:

Referees' comments:

Referee #1 (Remarks to the Author):

Knaut et al. report on entanglement distribution/generation between well-separated silicon vacancy colour centres in diamond. This work comprises quite a few key achievements besides the fundamental demonstration of entanglement formation over short (lab-scale) distances. E.g., by using quantum frequency conversion, the authors made their photons compatible with telecom fibre networks, which allowed them to distribute entanglement over 40 km fibre in the lab, and 35 km in a real-field scenario with deployed fibre. Additionally, entanglement was stored in long-lived nuclear spins up to 1 second, which has often been seen as a major challenge for silicon vacancy centres. These results also underline a good repeatability of the system (silicon vacancy centre in fibre-coupled nanophotonic waveguide cavity), which becomes quite important when considering scaling up the system.

Overall, these achievements stand as a key milestone towards long-distance quantum repeater networks, and I am thus convinced that the work will be of very high interest to a broad audience. I find the manuscript to be very well written and structured, the abstract is precise and accurate without any misleading overstatements, the methods and supplementary materials are exhaustive, and previous work is cited appropriately.

However, I have also a few critiques to mention, which I find important to be addressed before I can recommend a publication in Nature.

I start with my strongest critique, followed by a few suggestions and questions for clarification.

1.) While the authors manage to demonstrate entanglement generation in multiple scenarios, it must be noted that the achieved fidelities are probably not sufficient for promoting useful applications. I am aware that this work represents a “first-ever” demonstration, which may be even qualified as a hero effort. However, there seems to be some headroom for improvements. The authors correctly state quite a few strategies that “could be” implemented in the Conclusion section of the main text. However, non-experts (and even experts) may not be able to understand the potential impact of these improvements on fidelity and rates. I would feel happy if the authors could provide some rough estimates on the achievable improvements that can be reached, considering currently available knowledge and data (especially in the leading group of the authors). I think it would be important to understand how much improvements are directly achievable, and if there are some limitations of the scheme/system that prevent close-to-unity fidelities. Depending on the targeted application, very high fidelities in the range

of 99%+ may be required and it is important to understand if there are limitations when implementing improvements on the existing system. If such an analysis tends to become rather lengthy, it could be certainly shifted to the supplementary material.

In the following, I give my comments chronologically, as they appear in the Main Text and Supplementary Material.

2.) Page 1, line 22. I think that satellite quantum repeater networks could also be mentioned here for realising a global network infrastructure.

3.) Page 1, line 27/28: What is the purpose of mentioning “high-bandwidth interactions” here? To account/compensate naturally occurring spectral inhomogeneities within solid-state systems?

4.) Page 5, line 103: Please mention the inhomogeneous distribution as “plus/minus 50 GHz” to avoid misunderstanding (mention it in the same way as it is done in the Supplementary Material).

5.) Page 5, line 108: Please cite your group’s previous papers for frequency shifting and QFC.

6.) Page 7, line 156: After “using” error detection... I think that this mechanism is an excellent tool to improve the measured entanglement fidelity. However, non-experts may read the term “error detection” and naively assume a subsequent “error correction”. Obviously, the authors mention shortly before “discarding a measurement” (page 7, line 149), but I would be happy if discarding data was mentioned clearly also at later points in the manuscript. Maybe a term like “flag fault data” can be implemented, or some other wording.

7.) Page 8, line 192: The authors state that large-scale quantum networks “will have to use” existing fiber infrastructure.

 I believe that this statement is a bit too strong. On the short term, using existing infrastructure remains a cost-effective means to implement some demonstrator experiments (such as the excellent work presented here). However, imagining that a commercial scenario can be envisioned based on high enough entanglement fidelities and rates, it may be realistic to deploy dedicated quantum fibre networks. In this sense, I recommend a statement like “can strongly benefit from existing fiber infrastructure...”.

8.) Bibliography: Please check the references for correctness. I found incorrect page numbers for several references, such as 14, 36, 37, 38, and 46.

9.) Page 15, Fig. 3 caption.

Line 4: electric  electron.

Line 5: Integrated error detection  See comment 6.) above. Maybe extend the sentence by adding ...to flag faulty implementation of a PHONE gate.

Line 9: This may be a bit semantic, and I accept if the authors disagree. In my opinion, the authors demonstrate a “partial presence of entanglement” and not “entanglement” in the pure sense. However, it has become a standard in the community to speak about entanglement whenever the slightest signature of it is observable.

10.) Page 16, Fig. 4b). The error bars associated with the data points seem overestimated. If they were 1 std, I would expect 1-2 data points to be more than 1 error bar away from the modelled curve. Admittedly, the statistics is only based on 5 data points. However, I would recommend that the authors describe how the error bars have been obtained for this plot, maybe with a short text in the methods section. A reference to a description of the error bars should be mentioned at the end of the figure caption.

11.) Supplementary Material, page 8: The optical contrast error between electron-electron and nuclear-nuclear entanglement seems to vary between both experiments and the different nodes. Is there a systematic reason why the optical contrast error goes down for node A and up for node B? Or is this observation simply covered by the experimental uncertainties? In this case, why was it difficult to measure the optical contrast error more precisely?

Referee #2 (Remarks to the Author):

In this work, the authors demonstrate the generation of remote entanglement between two nodes consisting of SiVs with electron and nuclear spins and nanophotonic cavities. The results are impressive and the methods used appear justified and reliable.

Specifically, the telecommunication fiber connecting two nodes is tens of kilometers and the entanglement fidelity is ~ 0.7 , which achieves a quite high level compared to those demonstrated in other systems. In addition, when two nodes are connected directly via 20m long optical fiber, they show a high success rate $\sim 1\text{Hz}$ and a long storage time over 2s.

The work combining a series of advanced technologies is original and significant for realizing practical quantum networks.

Therefore, I think it is a promising paper and is suitable for Nature.

I have only a few minor comments.

1. The XX, YY, and ZZ correlations are measured for deriving the fidelity. However, in my opinion, it cannot be called Bell state tomography, which generally requires measuring a 4-by-4 density matrix. Also, the first letter of 'bell' is not capitalized in Fig. 2 and Fig. 3.

2. They only use fidelity to quantify the generated entanglement between the two nodes separated by tens of kilometers. How about the success rate and the storage time?

3. As shown in Fig. 1, the TDI measurement near node B can herald the generation of a Bell state. Thus the statement in lines 240-242 about spin-photon entanglement at node A is confusing.

4. The abbreviation 'TDI' is defined more than once.

Referee #3 (Remarks to the Author):

The capability of generating entanglement between two remote quantum memories is a key step towards the realization of quantum repeaters and quantum networks. In this paper, the authors experimentally demonstrated this capability. Importantly, they demonstrated entanglement generation between two quantum memory nodes that were connected via a 40 km long spools of fiber in lab and via a 35 km long fiber deployed in the Boston area urban environment respectively, and showed that the entanglement could survive for more than 1 second, long enough for the round-trip classical communication required for quantum teleportation.

To realize this capability, the authors use two Silicon Vacancy (SiV) centers in diamond that reside in two separate dilution refrigerators, each coupled to a fiber-interfaced diamond photonic crystal cavity in a spin-dependent way. The spin-dependent SiV-cavity coupling enabled the generation of spin-photon entanglement upon reflection of a single photon from the cavity. Thus, by sequentially scattering a single photon from each cavity, the two electron spins of the SiV center were projected into an entangled Bell state, heralded by the detection of the photon in the superposition basis. In addition, by using SiV centers with Si-29 that has a nuclear spin, they were able to generate entanglement between two remote nuclear spins that have much longer coherence time. Therefore, they were able to show the entanglement can be maintained for 1 s. Finally, the authors were able to show that the entanglement can be generated using both 40 km long optical fibers in lab, as well as 35 km long optical fibers deployed in the Boston area urban environment. This was achieved via frequency conversion of the photons from the SiV resonance of 737 nm to the telecom wavelength, as well as dynamical correction of the polarization drift happened fibers deployed in fields.

The results shown in this paper is built on the accumulation of multiple important breakthroughs in color centers, cavity QED, and quantum photonics, including nanophotonic cavity QED with large cooperativities, color centers with high-fidelity spin control and long coherence time, spin-photon entanglement, electron-nuclear spin entanglement, efficient quantum frequency conversion, etc. It also addresses many unique challenges associated with practical quantum repeater realization using solid-state spin systems, including frequency mismatch and polarization drift in fibers deployed in fields. Therefore, I believe this paper represents a major milestone in quantum optics, quantum information science, and quantum networks, and it will be of interest to a broad range of audience. Given the importance and significance of the results, I strongly encourage the publication of this paper in Nature in a timely way.

All statistics and error bars reported in the paper seem complete. I was impressed with the detailed fidelity and efficiency budget analysis.

I would like the authors to include the following details in the paper (likely in the supplement or methods) for reproducibility of their results.

1) Details of the cavity and device geometry, and the fabrication process. It seems that the cavity design and fabrication remain the same as their previous results. If this is the case, please clarify it explicitly in the paper.

2) Details of the data selection process. In one of the methods sections, the authors hinted that they were actively tracking the SiV's optical frequency and discarded the measurements where the spin-dependent reflectance contrast falls below a certain threshold. What is this threshold? How large fraction of data was discarded due to the reflectance contrast not meeting this threshold?

3) What is the contribution of the ionization laser to the reduction of the duty cycle? This was not mentioned in the discussion of the duty cycle.

Referee #4 (Remarks to the Author):

In emerging quantum technologies a major focus lies on the development of large scale quantum networks. Such networks hold great promise for ultimately secure communication and distributed quantum computing. To reach these goals, several experimental platforms are currently pushing their capabilities in order to move from lab-based demonstrations to real-world scenarios.

It is a particularly exciting time, as major building blocks for future large scale quantum networks are being demonstrated at a fast pace. This article is a beautiful example of this development. The authors employ cavity-coupled silicon-vacancy centers, and, to my knowledge, demonstrate for the first time entanglement distribution across two solid-state quantum network nodes with telecom photons. The authors further show entanglement between two nuclear spin memories with second long coherence times and entanglement distributed across ten's of km of fiber spools. This not being enough, the authors further use a real-world deployed fiber-loop in the Boston area and demonstrate successful heralding of quantum entanglement between the two network nodes. In particular, the last demonstration represents a major step over the state of the art of solid-state quantum network technology, paving the way for future distributed quantum networks based on colour center quantum spins.

The paper has a very clear story line, is written well and is highly accessible. The data and its description is presented in a clear manner and the manuscript stands on its own. In addition, the methods and the SuppMat add relevant information to understand certain aspects within more details. The results in this manuscript truly show a tour de force, joining previously developed ingredients as to perform an impressive demonstration of the potential of Silicon vacancy quantum network technology. I want to

congratulate the authors for this significant achievement and I am excited to recommend publication in Nature. I have, however, a couple of open points that, if addressed by the authors, would further improve the quality of the manuscript.

Major comments:

- I agree with the authors, that major building blocks for real-world quantum network applications are included in their sentence starting at line 33. Also, the work in this manuscript presents major breakthroughs along those lines. However, what is entirely missing in the manuscript is the aspect about scalability. Is the demonstrated approach extendable to more than two quantum network nodes? Can entanglement be stored in the memory qubits while new entanglement attempts are being performed? How would their system compare to state of the art solid-state spin based multinode quantum networks, see e.g. [10.1126/science.abg1919](https://doi.org/10.1126/science.abg1919). A particular strength of the demonstrated protocol lies in their single-photon heralding method, which does not require (additional) phase stability on the entanglement link. Would their serial architecture also work for multinode quantum networks, or would it be needed to switch to a parallel architecture (at the expense of losing the advantage of not needing fast active phase stabilization of the fiber link)? I see these questions of particular importance and recommend that the aspect of scalability is considered within the main manuscript.
- Φ^+_{nn} data: Throughout the main manuscript this data is omitted without any explanation. It is shown in the SuppMat, but also there without any explanation on the observed dependency on decoupling time and fiber length. It is important that the authors clarify in the main manuscript why they have decided to “hide” these data sets in the SuppMat. Also, can the authors comment on why the measured fidelity of data in Fig. S7 shows this peculiar shape (one would expect a gradual decrease with fiber length, while rather a maximum at 20km is observed)?

Minor comments:

- Line 70: can the authors comment why the cooperativity of node B is smaller by an order of magnitude compared to node A? Is this a designed target value, or something the authors do not have under control?
- Line 85: it would be good to already here give a reference to the methods as to allow the reader to understand the form of the resulting photon-electron Bell state.
- Figure 1:
 - o In (b) it would help to insert the symbols of line 65, such that the reader can immediately understand the ground state level structure (in addition to the colored spins)
 - o I recommend to insert the (line-of-sight) distance between the two SiV devices both in the Figure, as well in the text/caption.
- Figure 2:
 - o (a) for consistency it would be better to give a “+-” measurement symbol for the photon (instead of an “x” measurement)
 - o (b) in the data there are major deviations for the Φ^+ XX measurement from the ideal values – can the author give an explanation for the observed multiple-sigma asymmetry between “-X-X” and “XX”
 - o (b) can the authors add “theory correlators” to their data, i.e. the correlators that are expected from the independently measured performance of their system?

o (c) It is not clear what the filled curve is showing. Is it the expected fidelity as estimated from independently measured ingredients? I think “error-simulations” is not a clear term; do the authors mean a “model taking independently measured parameters/infidelities into account”?

o (c) I recommend to use a double x axis as to incorporate the employed “mean photon number” in addition to the success rate

- Figure 3:

o (a) for consistency it would be better to give a “+” measurement symbol for the photon (instead of an “x” measurement)

o (b) same comments as for Fig.2b

- Figure 4:

o (c) same comments as for Fig.2b

Author Rebuttals to Initial Comments:

Response to Referee Reports for "Entanglement of Nanophotonic Quantum Memory Nodes in a Telecom Network" [2023-09-16756] (Dated: December 20, 2023)

We would like to thank all four referees for the careful review of our manuscript and many useful comments and suggestions. In response, we have made a number of revisions, which, in our view, substantially improved the manuscript. Please find below our point-by-point responses to all referee comments with revisions indicated where appropriate. References are listed at the end of this document. Unless mentioned, the reference numbering is referring to the one in this document, not the one in the main manuscript. This document uses the following convention:

Original referee reports appear in black.

Responses to referee reports appear in blue.

Additions / changes to the manuscript appear in red.

REFEREE 1

Knaut et al. report on entanglement distribution/generation between well-separated silicon vacancy colour centres in diamond. This work comprises quite a few key achievements besides the fundamental demonstration of entanglement formation over short (lab-scale) distances. E.g., by using quantum frequency conversion, the authors made their photons compatible with telecom fibre networks, which allowed them to distribute entanglement over 40 km fibre in the lab, and 35 km in a real-field scenario with deployed fibre. Additionally, entanglement was stored in long-lived nuclear spins up to 1 second, which has often been seen as a major challenge for silicon vacancy centres. These results also underline a good repeatability of the system (silicon vacancy centre in fibre-coupled nanophotonic waveguide cavity), which becomes quite important when considering scaling up the system. Overall, these achievements stand as a key milestone towards long-distance quantum repeater networks, and I am thus convinced that the work will be of very high interest to a broad audience. I find the manuscript to be very well written and structured, the abstract is precise and accurate without any misleading overstatements, the methods and supplementary materials are exhaustive, and previous work is cited appropriately.

We thank the referee for their positive evaluation of our manuscript.

However, I have also a few critiques to mention, which I find important to be addressed before I can recommend a publication in Nature.

I start with my strongest critique, followed by a few suggestions and questions for clarification.

1. While the authors manage to demonstrate entanglement generation in multiple scenarios, it must be noted that the achieved fidelities are probably not sufficient for promoting useful applications. I am aware that this work represents a "first-ever" demonstration, which may be even qualified as a hero effort. However, there seems to be some headroom for improvements. The authors correctly state quite a few strategies that "could be" implemented in the Conclusion section of the main text. However, non-experts (and even experts) may not be able to understand the potential impact of these improvements on fidelity and rates. I would feel happy if the authors could provide some rough estimates on the achievable improvements that can be reached, considering currently available knowledge and data (especially in the leading group of the authors). I think it would be important to understand how much improvements are directly achievable, and if there are some limitations of the scheme/system that prevent close-to-unity fidelities. Depending on the targeted application, very high fidelities in the range of 99%+ may be required and it is important to understand if there are limitations when implementing improvements on the existing system. If such an analysis tends to become rather lengthy, it could be certainly shifted to the supplementary material.

We thank the referee for their review and agree that more specific numbers can be given for fidelity improvements to enhance the understanding of the limitations of our system and protocol. The error sources can be grouped in four categories: TDI locking error, optical contrast error, MW pulse error, and multi-photon error. Using realistically achievable lower-bound errors for all four error-sources, we estimate that we are able to generate electron-electron entanglement with fidelities of $\sim 95\%$ using the current experimental configuration. To reach fidelities of $> 99\%$, improved single photon sources (at the frequency and bandwidth of the SiV) would need

to be developed. However, for a range of quantum network applications, the requirements on entanglement fidelities are more relaxed. For example, quantum secured communication can be guaranteed at quantum bit error rates of below 11% [1]. Moreover, even for tasks such as connecting error-corrected quantum computers, fidelities in the range of ~ 0.9 might be sufficient [2]. Such fidelities are realistically attainable for our system using the above-mentioned improvements.

In terms of success rates, improved fiber-coupling efficiency and device cooperativity, and the introduction of strain tuning to remove the need to frequency conversion could improve the success rates of electron-electron entanglement generation to around 100 Hz. The rates could be further increased by using a parallel protocol, which would require phase locking of individual paths, or by employing multiple SiV-cavity systems per node in a multiplexed fashion. We thank the referee again for the question. We added the following estimation of achievable rates and fidelities in the "Conclusion" section of the main text:

Implementing the above improvements, electron-electron entanglement fidelities of ~ 0.95 with success rates of ~ 100 Hz could be achieved [3].

We furthermore added the following section "System Improvements" to the Supplementary Information:

The generated electronic Bell state error sources can be grouped into four categories: TDI locking error, optical contrast error, MW pulse error, and multi-photon error. There is no fundamental lower bound to the TDI locking error, and a straightforward improvement of the TDI design, including passive locking and environmental isolation through integration in a vacuum box, could reduce this error source to near-zero. The optical contrast error can be significantly suppressed as well. In principle, infinite contrast (and thus zero contrast error) can be achieved for any overcoupled SiV-cavity system with a cooperativity greater than one. In practice, SiV optical resonance diffusion and stray reflections can lower the contrast, though higher cooperativities mitigate these effects. With reasonable pre-selection and overcoupled SiV-cavity systems with cooperativities higher than 10, optical contrasts of 1:50 are achievable, which would reduce this error source to $\sim 4\%$ for $|\Phi_{ee}^+\rangle$ and $< 1\%$ for $|\Phi_{ee}^-\rangle$.

We further note that the SiV at node B was coupled to a nearby ^{13}C that caused a reduction in MW pulse fidelities. MW pulses are limited by spectral diffusion, spin-state decoherence, as well as MW driving-induced heating. Another constraint on MW pulses is the ^{29}Si state-dependent splitting of the electron MW transition [4]. This limits how short a MW pulse can be to selectively drive one transition to about 13 ns to apply C_nNOT_e gates, or requires very large Rabi frequencies ($\Omega \gg 33$ MHz) to effectively drive both transitions to apply an unconditional NOT_e gate. While MW gate errors of $\sim 0.1\%$ have previously been demonstrated [4], solid state microscopic environmental noise and frequency-dependent noise differs from emitter to emitter, imposing different error limits for different emitters. However, implementing pulse optimization techniques could allow reproducible and deterministic reduction of MW gate errors below 1%.

When using WCS as single photons, the error scales as $\sim \mu$, where μ is the average photon number of the WCS. To reach increasingly higher fidelities requires the use of increasingly lower photon numbers, which would cause a significant reduction in Bell state generation rate. If we replace the WCS with a single photon source, there would not be a need to sacrifice entanglement generation efficiency for fidelity. In this case, the multi-photon error would scale as the single photon source second order correlation at zero delay $\sim g^{(2)}(0)$. A state-of-the-art single photon source at the frequency and bandwidth of the SiV has previously shown to achieve $g^{(2)}(0) = 1.68\%$ [5]. Further improving this fidelity would require improved single photon sources at the frequency and bandwidth of the SiV. Compiling all these improvements, fidelities of around 0.95 could be achievable in the near future, with even higher fidelities reachable in the longer term with higher cooperativity SiV-cavity systems and improved single photon sources. Additionally to using a single photon source, the success rates of entanglement generation can further be improved by improving fiber coupling efficiency, with 95% coupling efficiency shown in previous work [6]. Higher cooperativity SiV-cavity systems could also enhance the cavity reflectance to 90% [6]. This could further boost the success rate by a factor of 7.7, yielding success rates of ~ 8 Hz. Access to strain tuning would remove the need for frequency shifting, further increasing the success rate by a factor of 13.5, resulting in an success rate rate of 100 Hz.

In the following, I give my comments chronologically, as they appear in the Main Text and Supplementary Material.

2. Page 1, line 22. I think that satellite quantum repeater networks could also be mentioned here for realising a global network infrastructure.

We thank the referee for this relevant comment. We agree that satellite-based quantum networks do offer an alternative approach to building large-scale quantum repeater networks, and added a mention as suggested (addition in bold).

Proposed architectures require quantum nodes containing multiple long-lived qubits that can collect, store, and process information communicated via photonic channels based on telecommunication (telecom) fibers **or satellite-based links**.

3. Page 1, line 27/28: What is the purpose of mentioning “high-bandwidth interactions” here? To account/compensate naturally occurring spectral inhomogeneities within solid-state systems?

We thank the referee for asking this clarifying question. In this context, “high-bandwidth interactions” is referring to the bandwidth of the photon being able to interact with the SiV. For the two cavity-QED systems used in this work, interactions with photons of bandwidths of up to 100 MHz were achieved, corresponding to 10 ns long photons. Based on the referee’s comment we decided that the use of the term “high-bandwidth” in this context might be confusing and decided to remove it”

Since photons and individual matter qubits interact weakly in free space [7], a promising approach to enhance the interaction between light and communication qubits is to utilize nanophotonic cavity quantum electrodynamic (QED) systems, where tight light confinement inside the nanostructure enables strong, ~~high-bandwidth~~ interactions between the photon and the communication qubit [8–11].

4. Page 5, line 103: Please mention the inhomogeneous distribution as “plus/minus 50 GHz” to avoid misunderstanding (mention it in the same way as it is done in the Supplementary Material).

We thank the referee for this suggestion, which we are happy to adopt. The main text has been changed to the following (addition in bold):

Since cavity-coupled ^{29}SiV centers possess an inhomogeneous distribution of optical transition frequencies of around ± 50 GHz centered around 406.640 THz (737.2 nm) [...]

5. Page 5, line 108: Please cite your group’s previous papers for frequency shifting and QFC.

We thank the referee for the suggestion and have added citations [12, 13] to line 108.

6. Page 7, line 156: After “using” error detection. . . I think that this mechanism is an excellent tool to improve the measured entanglement fidelity. However, non-experts may read the term “error detection” and naively assume a subsequent “error correction”. Obviously, the authors mention shortly before “discarding a measurement” (page 7, line 149), but I would be happy if discarding data was mentioned clearly also at later points in the manuscript. Maybe a term like “flag fault data” can be implemented, or some other wording.

We thank the referee for this clarifying comment. We have modified the main text as follows to point out the use of the electronic flag qubit (addition in bold):

After using error detection **by discarding measurements where the electronic flag qubit is measured in the $|\uparrow_e^A\rangle$ state**, the Bell state fidelity is $\mathcal{F}_{|\Phi_{nn}\rangle}^{\text{ED}} = 0.77(5)$, which is an improvement from the directly measured value of $\mathcal{F}_{|\Phi_{nn}\rangle}^{\text{raw}} = 0.64(5)$ without error detection.

7. Page 8, line 192: The authors state that large-scale quantum networks “will have to use” existing fiber infrastructure. → I believe that this statement is a bit too strong. On the short term, using existing infrastructure remains a cost-effective means to implement some demonstrator experiments (such as the excellent work presented here). However, imagining that a commercial scenario can be envisioned based on high enough entanglement fidelities and rates, it may be realistic to deploy dedicated quantum fibre networks. In this sense, I recommend a statement like “can strongly benefit from existing fiber infrastructure. . .”.

We thank the referee for this important comment. A fiber-based large-scale quantum network requires a deployed fiber network, but this fiber network does indeed not necessarily needs to be a preexisting one, but can be a newly deployed, dedicated one. We have adjusted the main text as follows to clarify this point (addition in bold):

In a practical setting, large-scale quantum networks ~~will have to use~~ **can strongly benefit from** existing fiber infrastructure to allow for long-distance entanglement distribution.

8. Bibliography: Please check the references for correctness. I found incorrect page numbers for several references, such as 14, 36, 37, 38, and 46.

We thank the referee for this observation, we have verified the correctness of the references and have corrected page numbers for [14], [36], [37], [38], and [46]. Furthermore, references [17], [19], [47], [50], and [51] got published between the submission of this manuscript and the first revision, so we change the references from the arXiv versions to the published versions. Note that the mentioned citations follow the numbering of the main manuscript.

9. Page 15, Fig. 3 caption.

- Line 4: electric \rightarrow electron, Line 5: Integrated error detection \rightarrow See comment 6.) above. Maybe extend the sentence by adding ... to flag faulty implementation of a PHONE gate.

We thank the referee for these useful suggestions, which we have incorporated into the following changes sentence to the main text (addition in bold):

Measurement of the ~~electric~~ electron spin qubits allows for integrated error detection by **flagging MW gate errors occurred during the PHONE gate.**

- Line 9: This may be a bit semantic, and I accept if the authors disagree. In my opinion, the authors demonstrate a “partial presence of entanglement” and not “entanglement” in the pure sense. However, it has become a standard in the community to speak about entanglement whenever the slightest signature of it is observable.

We thank the referee for this relevant comment. We indeed did follow the convention of the community to equate a Bell state fidelity of above 0.5 to an indication of an entangled state. This statement can be formalized as follows: We first define the Bell state fidelity for an arbitrary quantum state ρ with respect to the Bell states $|\Phi^\pm\rangle$ as $\mathcal{F}_\rho^{|\Phi^\pm\rangle} = \langle\Phi^\pm|\rho|\Phi^\pm\rangle$. We further define an entanglement witness operator $W = -|\Phi^\pm\rangle\langle\Phi^\pm| + \frac{1}{2}\mathbb{1}$. Measuring a negative value for $\langle W \rangle_\rho$ for any entanglement witness is certifying an entangled state [14]. Since $\mathcal{F}_\rho^{|\Phi^\pm\rangle} > 0.5 \Rightarrow \langle W \rangle_\rho < 0$, we can interpret a Bell state fidelity of above 0.5 as a certification for an entangled state. We thank the referee again for this question and hope that the referee agrees with our argument.

10. Page 16, Fig. 4b). The error bars associated with the data points seem overestimated. If they were 1 std, I would expect 1-2 data points to be more than 1 error bar away from the modelled curve. Admittedly, the statistics is only based on 5 data points. However, I would recommend that the authors describe how the error bars have been obtained for this plot, maybe with a short text in the methods section. A reference to a description of the error bars should be mentioned at the end of the figure caption.

We thank the referee for this observation. All error estimates of final Bell state fidelities shown in Fig. 2c, Fig. 3c and Fig. 4b have been obtained using the same standard method. The fact that all five datapoints in Fig 4b lie within one standard deviation of the solid model curve follows most likely from the low number of total observations. However, we agree with the referee that our manuscript would benefit from a description of the error calculation and have added the following section named “Bell State Fidelity and Error Calculation” to the Supplementary Information:

Both Bell state fidelity and the sample standard deviation of the Bell state fidelity can be obtained from the measured ZZ, XX, and YY correlators. The Bell state fidelity $\mathcal{F}_\rho^{|\Phi^\pm\rangle}$ of an arbitrary quantum state ρ with respect to the Bell states $|\Phi^\pm\rangle$ can be expressed as $\mathcal{F}_\rho^{|\Phi^\pm\rangle} = \frac{1}{2}P_{zz} + \frac{1}{4}P_{xx} + \frac{1}{4}P_{yy}$ [4], with

$$\begin{aligned} P_{zz} &\equiv p_{zz}^{00} + p_{zz}^{11} \\ P_{xx} &\equiv \begin{cases} p_{xx}^{01} + p_{xx}^{10} - p_{xx}^{00} - p_{xx}^{11}, & \text{for } |\Phi^-\rangle \\ p_{xx}^{00} + p_{xx}^{11} - p_{xx}^{01} - p_{xx}^{10}, & \text{for } |\Phi^+\rangle \end{cases} \\ P_{yy} &\equiv \begin{cases} p_{yy}^{00} + p_{yy}^{11} - p_{yy}^{01} - p_{yy}^{10}, & \text{for } |\Phi^-\rangle \\ p_{yy}^{01} + p_{yy}^{10} - p_{yy}^{00} - p_{yy}^{11}, & \text{for } |\Phi^+\rangle. \end{cases} \end{aligned}$$

Here, p_{nn}^{ij} , $nn \in \{zz, xx, yy\}$ describe the probabilities obtaining the various measurement outcomes $(i, j) \in \{0, 1\}^2$ for each measurement basis. Now, the sample variance $\sigma_{\mathcal{F}}^2$ of the Bell state fidelity can be expressed as $\sigma_{\mathcal{F}}^2 = \left(\frac{1}{2}\sigma_{zz}\right)^2 + \left(\frac{1}{4}\sigma_{xx}\right)^2 + \left(\frac{1}{4}\sigma_{yy}\right)^2$, with $\sigma_{nn \in \{zz, xx, yy\}}^2$ describing the sample variance of $P_{nn \in \{zz, xx, yy\}}$. Using the fact that $P_{nn \in \{zz, xx, yy\}}$ follows a binominal distribution, we can use the following expression of the sample

variance of a binomially distributed variable with success probability p : $\sigma^2(N, p) = \frac{p(1-p)}{N}$, where N is the sample size. Noting that for P_{zz} , $p = P_{zz}$, while for P_{xx} and P_{yy} , $p = \frac{P_{nn \in \{xx, yy\}} + 1}{2}$, we can finally express the sample standard deviation of the Bell state fidelity as:

$$\sigma_{\mathcal{F}} = \sqrt{\frac{1}{4} \frac{P_{zz}(1-P_{zz})}{N_{zz}} + \frac{1}{16} \frac{(1+P_{xx})(1-P_{xx})}{N_{xx}} + \frac{1}{16} \frac{(1+P_{yy})(1-P_{yy})}{N_{yy}}} \quad (1)$$

Here, $N_{nn \in \{zz, xx, yy\}}$ is the sample size for the measurements in the zz , xx , and yy basis.

11. Supplementary Material, page 8: The optical contrast error between electron-electron and nuclear-nuclear entanglement seems to vary between both experiments and the different nodes. Is there a systematic reason why the optical contrast error goes down for node A and up for node B? Or is this observation simply covered by the experimental uncertainties? In this case, why was it difficult to measure the optical contrast error more precisely?

We thank the referee for this careful observation. Due to the nuclear-state dependent reflectance [4], contrast errors do effect nucleus-nucleus entanglement generation more than electron-electron entanglement generation. In addition to this effect, changes in the solid-state environment impacting the spectral diffusion of the SiV and macroscopic shifts of the resonance frequency of the nanophotonic cavity change the observed magnitude of this error from experiment to experiment, increasing the uncertainty estimate of this measurement.

We again thank the referee for the very detailed review and believe that we could improve the manuscript by incorporating the referee's comments.

REFEREE 2

In this work, the authors demonstrate the generation of remote entanglement between two nodes consisting of SiVs with electron and nuclear spins and nanophotonic cavities. The results are impressive and the methods used appear justified and reliable.

Specifically, the telecommunication fiber connecting two nodes is tens of kilometers and the entanglement fidelity is ~ 0.7 , which achieves a quite high level compared to those demonstrated in other systems. In addition, when two nodes are connected directly via 20m long optical fiber, they show a high success rate ~ 1 Hz and a long storage time over 2 s.

The work combining a series of advanced technologies is original and significant for realizing practical quantum networks.

Therefore, I think it is a promising paper and is suitable for Nature.

We thank the referee for the concise summary of our work, the positive evaluation, and for recommending publication of our work in Nature.

I have only a few minor comments.

1. The XX , YY , and ZZ correlations are measured for deriving the fidelity. However, in my opinion, it cannot be called Bell state tomography, which generally requires measuring a 4-by-4 density matrix. Also, the first letter of 'bell' is not capitalized in Fig. 2 and Fig. 3.

We thank the referee for this accurate observation. The measurement results of the XX , YY , and ZZ correlations are indeed insufficient to fully describe the 4×4 density matrix ρ of the two-qubit state, only the absolute value $|\rho|$ could be reconstructed, see for instance [4]. We have changed "Bell state tomography" to "Bell state measurement" throughout the manuscript and in the figures to avoid any confusion. We also capitalized "Bell" in Fig. 2 and Fig. 3.

2. They only use fidelity to quantify the generated entanglement between the two nodes separated by tens of kilometers. How about the success rate and the storage time?

We thank the referee for this question. For the different fiber distances, rates between 8.2 mHz (0 km) and 0.23 mHz (35 km deployed) have been achieved. The quoted rates are averaged between $|\Phi_{nn}^+\rangle$ and $|\Phi_{nn}^-\rangle$. For more detailed information, please see Supplementary Information, table S6. The storage time was chosen to be 10 ms to allow for the classical signal traveling time between the nodes, which would be necessary assuming true space-like separated nodes.

3. As shown in Fig. 1, the TDI measurement near node B can herald the generation of a Bell state. Thus the statement in lines 240-242 about spin-photon entanglement at node A is confusing.

We thank the referee for this relevant comment. In the experiment described in the main text, we indeed perform a TDI measurement at node B, in order to herald the generation of a electronic or nuclear Bell state. The referenced lines 240-242 in the Methods section are describing the hypothetical TDI measurement after the $e - \gamma$ and PHONE gate being executed at a single node, similarly to previously presented work [4, 9]. We have added the following clarification in the Methods section "Spin-photon gates" for clarity:

The functionality of the two gates can be described in a single node configuration, where a TDI measurement is performed after the node.

4. The abbreviation 'TDI' is defined more than once.

We thank the referee for this comment. We have removed the redundant definition of "TDI".

We thank the referee again for the detailed review and the valuable suggestions and comments, which have improved the quality of the manuscript.

REFEREE 3

The capability of generating entanglement between two remote quantum memories is a key step towards the realization of quantum repeaters and quantum networks. In this paper, the authors experimentally demonstrated this capability. Importantly, they demonstrated entanglement generation between two quantum memory nodes that were connected via a 40 km long spools of fiber in lab and via a 35 km long fiber deployed in the Boston area urban environment respectively, and showed that the entanglement could survive for more than 1 second, long enough for the round-trip classical communication required for quantum teleportation.

To realize this capability, the authors use two Silicon Vacancy (SiV) centers in diamond that reside in two separate dilution refrigerators, each coupled to a fiber-interfaced diamond photonic crystal cavity in a spin-dependent way. The spin-dependent SiV-cavity coupling enabled the generation of spin-photon entanglement upon reflection of a single photon from the cavity. Thus, by sequentially scattering a single photon from each cavity, the two electron spins of the SiV center were projected into an entangled Bell state, heralded by the detection of the photon in the superposition basis. In addition, by using SiV centers with Si-29 that has a nuclear spin, they were able to generate entanglement between two remote nuclear spins that have much longer coherence time. Therefore, they were able to show the entanglement can be maintained for 1 s. Finally, the authors were able to show that the entanglement can be generated using both 40 km long optical fibers in lab, as well as 35 km long optical fibers deployed in the Boston area urban environment. This was achieved via frequency conversion of the photons from the SiV resonance of 737 nm to the telecom wavelength, as well as dynamical correction of the polarization drift happened fibers deployed in fields.

The results shown in this paper is built on the accumulation of multiple important breakthroughs in color centers, cavity QED, and quantum photonics, including nanophotonic cavity QED with large cooperativities, color centers with high-fidelity spin control and long coherence time, spin-photon entanglement, electron-nuclear spin entanglement, efficient quantum frequency conversion, etc. It also addresses many unique challenges associated with practical quantum repeater realization using solid-state spin systems, including frequency mismatch and polarization drift in fibers deployed in fields. Therefore, I believe this paper represents a major milestone in quantum optics, quantum information science, and quantum networks, and it will be of interest to a broad range of audience. Given the importance and significance of the results, I strongly encourage the publication of this paper in Nature in a timely way.

We thank the referee for the this thorough synopsis of our work, the positive evaluation, and the recommendation to publish our work in Nature.

All statistics and error bars reported in the paper seem complete. I was impressed with the detailed fidelity and efficiency budget analysis.

I would like the authors to include the following details in the paper (likely in the supplement or methods) for reproducibility of their results.

1. Details of the cavity and device geometry, and the fabrication process. It seems that the cavity design and fabrication remain the same as their previous results. If this is the case, please clarify it explicitly in the paper.

We thank the referee for this clarifying question. The device geometry and the fabrication process remains unchanged from previous experiments, details can be found in [5, 15]. We have added the following sentence in the Supplementary Information, section "Cavity-QED Parameters".

The design and fabrication of the nanophotonic cavity is described in [5, 15].

2. Details of the data selection process. In one of the methods sections, the authors hinted that they were actively tracking the SiV's optical frequency and discarded the measurements where the spin-dependent reflectance contrast falls below a certain threshold. What is this threshold? How large fraction of data was discarded due to the reflectance contrast not meeting this threshold?

We thank the referee for this important question. The optical contrast of the SiVs is fluctuating due to spectral diffusion of the emitter and slow, macroscopic drifts of the resonance frequency of the nanophotonic cavity. On average, this contrast threshold for the SiV in node A was set to 1:16, while for the SiV in node B it was set to 1:8. These thresholds resulted in an average rejection of 23% of datapoints per dataset. We have added the following sentence to the Methods section "Success rates":

On average, this contrast threshold for the SiV in node A was set to 1:16, while for the SiV in node B it was set to 1:8. These thresholds resulted in an average rejection of 23% of datapoints per dataset.

3. What is the contribution of the ionization laser to the reduction of the duty cycle? This was not mentioned in the discussion of the duty cycle.

We thank the referee for this comment. Indeed, the application of the green ionization laser results in a reduction of the duty cycle D . The exact quantification of this reduction is non-trivial for our experimental configuration. Specifically, we have both a hardware-timed de-ionization procedure directly triggered by the FPGA of our AWG, as well as a software-timed procedure compensating for spectral diffusion, both of which are running interleaved with each other. Thus, accurately determining the impact of one of the two effects on the duty-cycle is challenging. Nevertheless, we agree with the referee that it is important to mention the fact that the application of the ionization laser reduces the duty cycle. We have added the following sentence to the "Success rates" section of the Methods:

Furthermore, the application of the green ionization laser to reset the SiV's charge state additionally reduces D .

We would like to thank the referee again for drafting this thorough review, and many useful comments and suggestions.

REFEREE 4

In emerging quantum technologies a major focus lies on the development of large scale quantum networks. Such networks hold great promise for ultimately secure communication and distributed quantum computing. To reach these goals, several experimental platforms are currently pushing their capabilities in order to move from lab-based demonstrations to real-world scenarios. It is a particular exciting time, as major building blocks for future large scale quantum networks are being demonstrated at a fast pace. This article is a beautiful example of this development. The authors employ cavity-coupled silicon-vacancy centers, and, to my knowledge, demonstrate for the first time entanglement distribution across two solid-state quantum network nodes with telecom photons. The authors further show entanglement between two nuclear spin memories with second long coherence times and entanglement distributed

across ten's of km of fiber spools. This not being enough, the authors further use a real-world deployed fiber-loop in the Boston area and demonstrate successful heralding of quantum entanglement between the two network nodes. In particular, the last demonstration represents a major step over the state of the art of solid-state quantum network technology, paving the way for future distributed quantum networks based on colour center quantum spins.

The paper has a very clear story line, is written well and is highly accessible. The data and its description is presented in a clear manner and the manuscript stands on its own. In addition, the methods and the SuppMat add relevant information to understand certain aspects within more details. The results in this manuscript truly show a tour de force, joining previously developed ingredients as to perform an impressive demonstration of the potential of Silicon vacancy quantum network technology. I want to congratulate the authors for this significant achievement and I am excited to recommend publication in Nature. I have, however, a couple of open points that, if addressed by the authors, would further improve the quality of the manuscript.

We thank the referee for this comprehensive summary and assessment of our work, the positive feedback, and the recommendation to publish our work in Nature.

Major comments:

1. I agree with the authors, that major building blocks for real-world quantum network applications are included in their sentence starting at line 33. Also, the work in this manuscript presents major breakthroughs along those lines. However, what is entirely missing in the manuscript is the aspect about scalability. Is the demonstrated approach extendable to more than two quantum network nodes? Can entanglement be stored in the memory qubits while new entanglement attempts are being performed? How would their system compare to state of the art solid-state spin based multinode quantum networks, see e.g. 10.1126/science.abg1919. A particular strength of the demonstrated protocol lies in their single-photon heralding method, which does not require (additional) phase stability on the entanglement link. Would their serial architecture also work for multinode quantum networks, or would it be needed to switch to a parallel architecture (at the expense of losing the advantage of not needing fast active phase stabilization of the fiber link)? I see these questions of particular importance and recommend that the aspect of scalability is considered within the main manuscript.

We thank the referee for this very relevant comment, and agree that the manuscript can be improved by addressing the question of system scalability. The demonstrated serial entanglement generation protocol can be extended to multiple nodes using different approaches. The two-node configuration could be extended to multiple nodes connected serially to directly generate multipartite entanglement, while the entanglement generation technique and nanophotonic cavity properties could remain unchanged. While this configuration is conceptually simplest, it is subject to added loss due to the additional frequency conversion and insertion loss of the added quantum network nodes. A more flexible network configuration could be obtained by employing a switch network between the nodes, which could allow for the generation of bipartite entanglement between two arbitrary nodes in the network. In order to extend this pairwise entanglement to multipartite entanglement as for instance demonstrated in [16], entanglement needs to be stored while new entanglement attempts are performed. Using the amplitude-based readout-technique employed in this work, the ^{29}Si nuclear spin experiences dephasing from the laser during readout of the electron spin, which would limit the number of possible entanglement attempts [4]. The use of phase-based readout, as previously demonstrated in [4], can increase the number of possible entanglement attempts, while swapping quantum information onto weakly coupled ^{13}C -spins extending this number further. Combining these improvements with the efficiency enhancements mentioned in the main text, we believe that our platform and serial architecture is well suited for scaling up to multi-node networks. We thank the referee again for this important question, and have added the following to the "Two-node quantum network using integrated nanophotonic systems" section of the main text (line 101):

Furthermore, extending the number of network nodes to more than two can be achieved by either connecting more than two nodes in series, or by employing a switch network between multiple nodes to generate pairwise connectivity.

We furthermore added the following to the "Conclusion" section of the main text (addition in bold):

Finally, the number of accessible qubits could be increased by addressing weakly coupled ^{13}C spins [17], **allowing for more flexible multi-node network configurations.**

2. $|\Phi_{nn}^+\rangle$ data: Throughout the main manuscript this data is omitted without any explanation. It is shown in the SuppMat, but also there without any explanation on the observed dependency on decoupling time and fiber length. It is important that the authors clarify in the main manuscript why they have decided to "hide" these data sets in the SuppMat. Also, can the authors comment on why the measured fidelity of data in Fig. S7

shows this peculiar shape (one would expect a gradual decrease with fiber length, while rather a maximum at 20km is observed)?

We thank the referee for this important question. As elaborated in the section "Contrast Error Distillation" in the Methods, the errors due to a finite reflectance contrast of the SiV-cavity system accumulate preferentially in the $|\Phi_{nn}^+\rangle$ state. Hence, selecting for heralding events for the $|\Phi_{nn}^-\rangle$ state allows for a higher Bell state fidelity, which is the main reason why the main manuscript focused on displaying the data for $|\Phi_{nn}^-\rangle$. Moreover, errors due to the finite reflectance contrast are varying over time due to spectral diffusion of the SiV, macroscopic shifts of the resonance frequency of the nanophotonic cavity, and the laser's frequency stability. For experiments with lower success rates (as for example the experiments described in Fig. 3 and Fig. 4), the magnitude of the contrast error can vary considerably from datapoint to datapoint. Due to this slowly varying contrast error, the dependence of the Bell state fidelities of the $|\Phi_{nn}^+\rangle$ state shown in Fig. S7 can produce features such as the peak in fidelity at 20 km fiber distance the referee correctly pointed out. We thank the referee for asking this very relevant question and agree with the referee that the main manuscript can be improved by adding clarification. We have added the following sentence to the main manuscript:

Similar to $|\Phi_{ee}^+\rangle$, the generated $|\Phi_{nn}^+\rangle$ state accumulates errors due to imperfect reflectance contrast, see [3] for more information.

We furthermore added this clarification to the "Entanglement Results Additional Data" section in the Supplementary Information:

Errors due to imperfect reflectance contrast preferentially affect the $|\Phi_{nn}^+\rangle$ state, see Section III. The magnitude of this type of error slowly varies over time due to spectral diffusion of the SiV, macroscopic shifts of the resonance frequency of the nanophotonic cavity, and the laser's frequency stability. These error sources contribute to the dependency of the Bell state fidelities on the fiber length shown in Fig. S7.

Minor comments:

2. Line 70: can the authors comment why the cooperativity of node B is smaller by an order of magnitude compared to node A? Is this a designed target value, or something the authors do not have under control?

We thank the referee for this clarifying question. While both nodes have been designed with a similar target cooperativity, the realized cooperativity did deviate from the target cooperativities mainly due to two factors: First, the single photon Rabi frequency g , which is related to the overlap between the SiV's electron orbitals and the mode-maximum of the nanophotonic cavity, can differ from device to device. This can be due, for instance, to the inherently probabilistic implantation depth of the silicon ions, which are used to create SiV centers. Furthermore, the total linewidth κ_{tot} of the nanophotonic cavity can differ from the design value due to slight run-to-run variations of process parameters in the fabrication process, like for instance slight thickness variation of the electron-beam resist. These effects taken together result in deviations of the realized cooperativities from the target cooperativities. More information on the cavity design and fabrication process can be found in [5, 15].

3. Line 85: it would be good to already here give a reference to the methods as to allow the reader to understand the form of the resulting photon-electron Bell state.

We thank the referee for this suggestion, and have added a reference to the Methods:

The resulting photon-electron Bell state can be described as $|\text{Photon, SiV A}\rangle = (|e \downarrow_e^A\rangle + |l \uparrow_e^A\rangle)/\sqrt{2}$ [18].

4. Figure 1.

- In (b) it would help to insert the symbols of line 65, such that the reader can immediately understand the ground state level structure (in addition to the colored spins)

We thank the referee for this suggestion and agree that adding the ket-symbols will facilitate the interpretation of the SiV's level structure. We have adopted this suggestion in the new figure draft.

- I recommend to insert the (line-of-sight) distance between the two SiV devices both in the Figure, as well in the text/caption.

We thank the referee for this suggestion, and agree that the line-of-sight distance between the two network nodes of 6 m should be mentioned. Adding the line-of-sight visually in the Figure would risk that the line-of-sight distance is confused with the in-fiber-distance. For this reason, we hope the referee agrees that it

is acceptable not to include the line-of-sight distance visually in Figure 1. To clearly state the line-of-sight distance, we have added the following to the caption of Figure 1:

The line-of-sight distance between the two SiVs is 6 m.

5. Figure 2.

- (a) for consistency it would be better to give a “+/-“ measurement symbol for the photon (instead of an “x” measurement)

We thank the referee for this suggestion, and agree that consistency can be improved by using the “+/-“ measurement symbol. For Fig. 2a and Fig. 3a., we have followed this suggestion in the new figure drafts.

- (b) in the data there are major deviations for the Φ^+ XX measurement from the ideal values – can the author give an explanation for the observed multiple-sigma asymmetry between “-X-X” and “XX”.

We thank the referee for asking this clarifying question. Using the theoretical model predicting the correlators based on independently measured performance parameters, we can gain insight into the cause for the mentioned asymmetry of the “-X+X” and “+X-X” measurement. Specifically, this asymmetry can be explained by the error due to finite reflectance contrast of the cavity-QED system. As described in the Methods section, this error mainly affects the $|\Phi_{ee}^+\rangle$ state, which is why the described asymmetry is more pronounced in the $|\Phi_{ee}^+\rangle$ state than in the $|\Phi_{ee}^-\rangle$ state. As can be seen by the added theory correlators (see answer to the next question), which takes the contrast error into account, the model does successfully replicate the asymmetry.

- (b) can the authors add “theory correlators” to their data, i.e. the correlators that are expected from the independently measured performance of their system?

We thank the referee for this useful suggestion, which we are happy to implement. For Fig. 2b, Fig. 3b and Fig. 4c, we have replaced the optimal values of the correlators by ones predicted by simulations using independently measured performance parameters of our system. For Fig. 2-4, we have adjusted the figure caption as follows:

~~Dashed bars show ideal values.~~ Dashed bars show correlations predicted by a theoretical model using independently measured performance parameters of our system.

- (c) It is not clear what the filled curve is showing. Is it the expected fidelity as estimated from independently measured ingredients? I think “error-simulations” is not a clear term; do the authors mean a “model taking independently measured parameters/infidelities into account”?

We thank the referee for this clarifying question. The referee’s interpretation is correct. The curve is showing predictions from a theory model, taking independently measured parameters into account. For each input parameter, a suitable parameter range centered around the measured value has been assumed. These values are reproduced in Table S1. We agree that the term “error-simulation” is not clearly defined, and following the referee’s suggestion have changed the caption as follows:

~~Filled curves show results of error-simulations.~~ Filled curves show predictions by a theory model using independently measured performance parameters of our system [3].

- (c) I recommend to use a double x axis as to incorporate the employed “mean photon number” in addition to the success rate

We thank the referee for this excellent suggestion, which we incorporated into the new figure draft.

6. Figure 3.

- (a) for consistency it would be better to give a “+/-“ measurement symbol for the photon (instead of an “x” measurement)

Please see answer to comment re. Fig. 2a.

- (b) same comments as for Fig.2b

We thank the referee for the suggestion to add correlator values predicted by a theoretical model using independently measured performance parameters of our system, which we happily adopted. The model

also can be used to partially explain the observed asymmetry between the "-X+X" and "+X-X" correlator. Specifically, the infidelity of the read-out of the nuclear states causes a biasing of the "-X+X / +X-X" and the "+Y+Y / -Y-Y" correlators. This error source is discussed in section "Fidelity Budgets" in the Supplementary Information. While the model successfully predicts the asymmetry of the "+Y+Y / -Y-Y" correlators, the asymmetric of the "-X+X / +X-X" is only partially accounted for. We believe that this is partly due to slow fluctuations of the MW transition frequencies due to ODMR-diffusion and measurement uncertainty, which especially for long experiments such as the ones presented in Fig. 3 and Fig. 4 can result a mismatch of the instantaneous MW pulse fidelities and the fidelities assumed by the model. We thank the referee again for this important observation.

7. Figure 4.

- (c) same comments as for Fig.2b
Please see answer to comment re. Fig. 3b.

We again thank the referee for this thorough review. We believe that we could improve the manuscript and figures by incorporating the referee's suggestions.

FURTHER CHANGES

Title: Change of "Telecommunication" to "Telecom" to comply with length limitation (addition in bold).
~~Entanglement of Nanophotonic Quantum Memory Nodes in a Telecommunication-~~ **Telecom** Network

Abstract: Added abbreviation of "Telecommunication" (addition in bold).

Here, we demonstrate a two-node quantum network composed of multi-qubit registers based on silicon-vacancy (SiV) centers in nanophotonic diamond cavities integrated with a telecommunication (**telecom**) fiber network. [...] By integrating efficient bi-directional quantum frequency conversion of photonic communication qubits to ~~telecommunication~~ **telecom** frequencies (1350 nm), [...]

Line 24: Correction of typo.

Since photons and individual matter qubits interact ~~weekly~~ weakly in free space, [...]

Figure 3, caption: Motivated by referee 4's comment regarding Fig. 2c., we have changed the caption of Figure 3 as follows:

~~Filled curves show results of error simulations.~~ Filled curves show predictions by a theory model using independently measured performance parameters of our system [3].

-
- [1] Lütkenhaus, N. Estimates for practical quantum cryptography. *Phys. Rev. A* **59**, 3301–3319 (1999).
 - [2] Ramette, J., Sinclair, J., Breuckmann, N. P. & Vuletić, V. Fault-tolerant connection of error-corrected qubits with noisy links. arXiv:2302.01296 (2023).
 - [3] See Supplementary Information for further clarification and discussion.
 - [4] Stas, P.-J. *et al.* Robust multi-qubit quantum network node with integrated error detection. *Science* **378**, 557–560 (2022).
 - [5] Knall, E. *et al.* Efficient source of shaped single photons based on an integrated diamond nanophotonic system. *Physical Review Letters* **129**, 053603 (2022).
 - [6] Bhaskar, M. K. *et al.* Experimental demonstration of memory-enhanced quantum communication. *Nature* **580**, 60–64 (2020).
 - [7] Reiserer, A. & Rempe, G. Cavity-based quantum networks with single atoms and optical photons. *Rev. Mod. Phys.* **87**, 1379–1418 (2015).
 - [8] Sipahigil, A. *et al.* An integrated diamond nanophotonics platform for quantum-optical networks. *Science* **354**, 847–850 (2016).
 - [9] Nguyen, C. T. *et al.* Quantum network nodes based on diamond qubits with an efficient nanophotonic interface. *Physical Review Letters* **123**, 183602 (2019).
 - [10] Ourari, S. *et al.* Indistinguishable telecom band photons from a single erbium ion in the solid state. *Nature* **620**, 977–981 (2023).

- [11] Ruskuc, A., Wu, C.-J., Rochman, J., Choi, J. & Faraon, A. Nuclear spin-wave quantum register for a solid-state qubit. *Nature* **602**, 408–413 (2022).
- [12] Bersin, E. *et al.* Telecom networking with a diamond quantum memory. arXiv:2307.08619 (2023).
- [13] Bersin, E. *et al.* Development of a boston-area 50-km fiber quantum network testbed. arXiv:2307.15696 (2023).
- [14] Gühne, O. & Tóth, G. Entanglement detection. *Physics Reports* **474**, 1–75 (2009).
- [15] Nguyen, C. T. *et al.* An integrated nanophotonic quantum register based on silicon-vacancy spins in diamond. *Physical Review B* **100**, 165428 (2019).
- [16] Pompili, M. *et al.* Realization of a multinode quantum network of remote solid-state qubits. *Science* **372**, 259–264 (2021).
- [17] Bradley, C. E. *et al.* A ten-qubit solid-state spin register with quantum memory up to one minute. *Physical Review X* **9**, 031045 (2019).
- [18] See Methods for further clarification and discussion.

Reviewer Reports on the First Revision:

Referees' comments:

Referee #1 (Remarks to the Author):

Knaut et al. have addressed all my comments (very) appropriately, such that I am very happy to recommend the paper for publication as it is now.

Referee #2 (Remarks to the Author):

The authors have addressed all my comments and other referees' satisfactorily and I recommend the present version for publication in Nature.

Referee #3 (Remarks to the Author):

The authors have addressed all my questions. I support the publication of the paper in Nature.

Referee #4 (Remarks to the Author):

The authors have properly taken all my comments into account, as well gave very detailed and sufficient answers to all the question I have raised. In particular, their vision for scalability and the author's choice of data is now more clearly understandable. Further, adding correlators predicted by simulations based on independently measured performance parameters eases the evaluation of the presented data. In addition, the author's detailed response to the comments of the other referees definitely helped to further improve the quality of the manuscript. I am happy with the improved manuscript and supplemental material, and can recommend publication of this manuscript as it is.